# TreeSMOTE: Structure-Aware Data Augmentation for Imbalanced Tabular Learning

## Abstract

Class imbalance has been a critical bottleneck in classification problems, undermining a classifier's identification of minority instances. Data augmentation provides an effective solution by oversampling the minority. Extant methods often generate samples through duplication, perturbation, or interpolation, largely relying on the assumption of local smoothness of the data space to ensure synthetic data reliability. Alternatively, generative models are leveraged for data learning and synthesis. However, both approaches encounter significant limitations in tabular data, primarily due to data heterogeneity and smaller sizes. Specifically, in tabular data with heterogeneous (e.g., continuous, categorical, and ordinal) features, local smoothness may not hold, and thus spatial proximity may fail to capture the underlying data distribution, resulting in noisy or misleading synthetic instances. Moreover, generative models can overfit (memorize) smaller tabular data and require and may require computationally expensive model training and tuning. To bridge the gap in tailored augmentation for imbalanced tabular classification, we propose TreeSMOTE, a structure-aware oversampling framework, as an out-of-the-box tool for imbalanced tabular classification. It shifts the focus from the feature space to a tree-induced decision space. We quantify the similarity between samples based on the depth of their lowest common ancestor in decision paths from trees and synthesize label-consistent instances based on structurally similar samples. TreeSMOTE readily combines with and improves classification models by increasing data diversity and reducing label contamination. Extensive experiments on large-scale imbalanced tabular learning benchmarks demonstrate that TreeSMOTE achieves competitive performance compared to popular oversampling methods and generative models under the default configuration, circumventing the need for expensive hyperparameter tuning or GPU acceleration. In addition, TreeSMOTE performs comparably to and can integrate with and further improve state-of-the-art undersampling ensemble approaches.

## 1 Introduction

Tabular data is one of the most prevalent data formats in real-world applications, such as financial transactions for fraud detection (Kennedy et al., 2025), electronic health records for medical diagnosis (Borisov et al., 2021), and industrial logs for anomaly detection (Ye et al., 2025). Class imbalance is a pervasive issue in tabular datasets, where events of interest occur with significantly lower frequency than majority-class instances (Xiao et al., 2021). Class imbalance poses substantial challenges for classification models, leading to predictions biased toward the majority class and poor recognition of the minority that is often of more practical importance.

A common strategy to mitigate class imbalance is data-level rebalancing, including undersampling and oversampling. Undersampling reduces the number of majority-class instances to balance the dataset, but it may discard informative samples and degrade classification performance (Liu et al., 2025). To preserve the information of the majority, oversampling methods that duplicate or generate synthetic samples for the minority class are widely adopted. Modern generative models can be used, like variational autoencoders (VAE), generative adversarial nets, and diffusion models can be adapted to tubular data synthesis (Xu et al., 2019; Adiputra & Wanchai, 2024; Ren et al., 2025). However, the deep generative models may

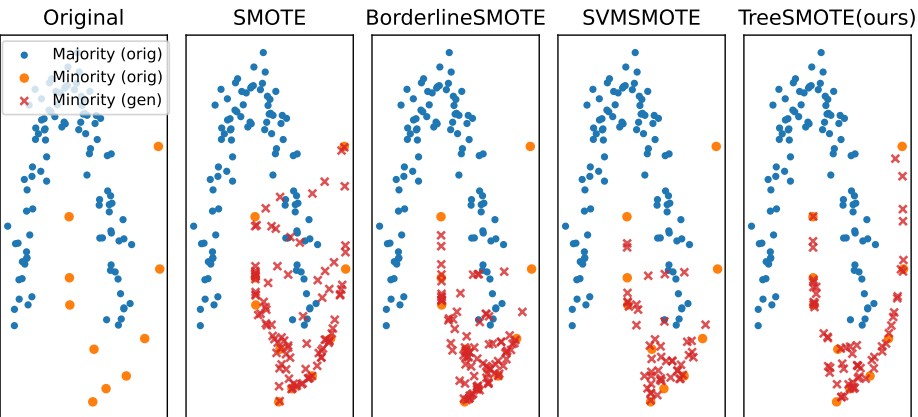

Figure 1: Minority-class augmentation by SMOTE-based methods. The first panel depicts the original data consisting of two crescent moons, with the upper moon (blue dots) being the majority and the lower (orange dots) being the minority. The other panels illustrate synthetic minority instances (red crosses) via SMOTE, BorderlineSMOTE, SVMSMOTE, and TreeSMOTE, respectively. However, TreeSMOTE generates instances for the minority in its region without violating class boundaries.

overfit (memorize) smaller-sized tabular data (e.g., van den Burg & Williams, 2021; Baptista et al., 2025), undermining the synthetic data diversity, and may require expensive model training, hyperparameter tuning, and GPU acceleration, creating adoption hurdles for end users.

Without being constrained by the data size, the Synthetic Minority Oversampling Technique (SMOTE) (Chawla et al., 2002) and its variants—such as BorderlineSMOTE (Han et al., 2005) and SVMSMOTE (Nguyen et al., 2011)—are widely adopted for augmenting minority classes. These methods synthesize new instances by interpolating between neighboring samples within the feature space. However, extant SMOTE methods largely relies on local smoothness of the data space between neighboring data points, a premise that is often violated in tabular datasets with heterogeneous (e.g., continuous, categorical, and ordinal) features (Wang et al., 2006). Consequently, the minority data synthesis introduces noisy or misleading instances and degrades classification performance (Fernández et al., 2018), because relying on spatial proximity (e.g., quantified by Euclidean distance) for neighbor selection fails to preserve the underlying data distribution (Feng et al., 2022; Dai et al., 2023).

To resolve this, we shift the perspective from spatial proximity to structural similarity by leveraging tree-induced decisions that partition the feature space into hierarchical, label-homogeneous regions, where local smoothness is better preserved. Samples that share a deeper decision path exhibit a higher structural similarity that transcends their raw feature disparities. We propose *TreeSMOTE*, a structure-aware oversampling framework that constrains data synthesis to decision-consistent regions. To further ensure data quality, a consistency filtering step is introduced to prune unreliable synthetic data with negligible computational overhead. Prioritizing structural similarity over spatial proximity, TreeSMOTE preserves the underlying data distribution and reduces label contamination.

In Figure 1, we illustrate the advantage of TreeSMOTE over SMOTE and its variants, BorderlineSMOTE and SVMSMOTE. These approaches generate minority-class instances violating class regions (i.e., red crosses and blue dots overlap), thereby distorting the original data manifold and undermining downstream classification. In contrast, TreeSMOTE identifies structurally similar neighbors that do not cross class boundaries, ensuring that synthetic samples remain within valid minority regions and preserve the data distribution.

Since tree models are inherently capable of processing nominal features without data transformation, TreeSMOTE is tailored for tabular data, leveraging tree-induced decisions to capture complex feature interdependence and heterogeneous data types—both are critical characteristics of tabular data that may be absent in unstructured data formats. Unlike oversampling that treats features as independent axes in a Euclidean space, TreeSMOTE captures the underlying conditional logic of the data. By measuring how

samples traverse hierarchical decision paths, our approach identifies structural similarities that respect class boundaries and feature interactions. We propose TreeSMOTE as an effective out-of-the-box tool for tabular data augmentation, which achieves highly competitive performance across varying imbalance ratios under a default configuration without expensive model selection. We make three primary contributions:

- Structural similarity via decision paths. We define tree-induced *structural similarity* and *structural gap* based on the depth of the lowest common ancestor in decision paths, offering a distinct and more effective alternative to Euclidean distance for finding proximal neighbors.

- Consistency filtering for improved reliability. The quality of the generated data is further enhanced through a consistency filtering step, which incurs no additional model training.

- Extensive empirical validation and competitive performance across diverse settings. Comparison against extensive baselines under default configurations and the advanced Self-paced Ensemble validates TreeSMOTE as a practical data augmentation tool for imbalanced tabular classification.

## 2 Related Literature

Extensive efforts have been directed toward addressing the challenges of class imbalance. We categorize the extant literature into three primary streams: optimization-based approaches, representation learning, and data-level rebalancing. The proposed TreeSMOTE contributes to data-level rebalancing and can be seamlessly combined with models in the other two streams. We discuss each stream and our contribution to the literature.

### 2.1 Optimization-Based Approaches

Optimization-based methods do not rely on classifier specification but on training schemes. They address class imbalance that often forces classifiers to favor the majority and overlook the minority (Ye et al., 2024). Rather than optimizing the classifier for an overall performance, these approaches prioritize accuracy across specific groups of interest. A significant body of work adopts distributionally robust optimization (DRO) to improve worst-case performance. For instance, Group-DRO (Sagawa et al., 2019) minimizes the maximum loss across groups rather than the global average. Subsequent research has further refined this by emphasizing the minority during training (Goel et al., 2020; Levy et al., 2020; Sagawa et al., 2020; Zhang et al., 2021; Deng et al., 2023).

Two-stage training has been developed to automatically identify the minority in a data-agnostic manner, minimizing the impacts of group labeling. In the first stage of training, they identify hard-to-classify instances (Nam et al., 2020; Liu et al., 2021; Yenamandra et al., 2023). In the second stage, they focus training on these instances to let the classifier better learn their distributions (Du et al., 2023; Moayeri et al., 2023; Yang et al., 2024). Our proposed TreeSMOTE framework complements existing approaches in this stream through seamlessly integrating with them and providing diversed supplementary samples, thereby facilitating a more robust training process.

### 2.2 Representation Learning

Representation learning in the presence of class imbalance focuses on capturing invariant features that remain consistently predictive across both the majority and minority (Sun et al., 2021; Veitch et al., 2021; Eastwood et al., 2023) and reducing the influence of spurious correlation between features and labels arising from biased data. Recent studies (Rosenfeld et al., 2022; Zhong et al., 2024) suggest that standard empirical risk minimization learns both spurious and invariant feature representations. This implies that prediction accuracy can be improved by highlighting invariant feature representations and downplaying the spurious.

Building on this, several techniques involve retraining the last layer of a neural network on small, held-out datasets where spurious correlations are broken (Kirichenko et al., 2023; Izmailov et al., 2022). They can be combined with automatic minority identification, such as using disagreements between regular empirical

risk minimization and early-stopping (Chen et al., 2023; LaBonte et al., 2023). In addition, regularization has been used to minimize the classifier's learning of spurious correlation. Specifically, using nuclear-norm penalty (Shi et al., 2024) or both nuclear- and Frobenius-norm penalties (Wen et al., 2025) exhibits similar properties to LASSO or elastic net to emphasize invariant feature representations.

## 2.3 Data-Level Rebalancing

Data-level interventions balance classes before training a classifier. This stream of work can be categorized into oversampling and undersampling. Oversampling, or data augmentation, seeks to expand the minority class by generating new instances rather than simple duplication. As one of the most popular augmentation schemes, SMOTE (Chawla et al., 2002) generates synthetic samples via interpolation between minority samples. To address potential noise and overlapping class boundaries, variants such as BorderlineSMOTE (Han et al., 2005) and SVMSMOTE (Nguyen et al., 2011) focus on generation in critical decision regions or support vectors, and Batista et al. (2004) propose to remove ambiguous instances using Tomek links or nearest neighbor rules. For tabular data generation, generative models can be leveraged, like variational autoencoders, generative adversarial nets, and diffusion models (Xu et al., 2019; Adiputra & Wanchai, 2024; Ren et al., 2025), but their training necessitate larger datasets and often impose significant computational demands.

Undersampling adjusts class proportions through removing majority instances but risks discarding informative samples. To address this limitation, ensemble-based approaches that combine base learners trained on undersampled, balanced data have become a prominent solution (Zhao et al., 2023). For example, diversity can be enhanced by training multiple classifiers on distinct undersampled majority subsets (Barandela et al., 2003), and performance is further improved by gradually focusing on hard-to-classify instances (Liu et al., 2008). Self-Paced Ensemble (Liu et al., 2020) considers classification difficulty by assigning higher sampling weights to challenging majority-class instances and has achieved state-of-the-art performance. Other representative methods, such as RUSBoost (Seiffert et al., 2009) and Balanced Random Forest (Khoshgoftaar et al., 2007), combine undersampling with boosting or bootstrap aggregation to enhance minority-class recall.

The proposed TreeSMOTE contributes to data-level rebalancing approaches. As a data augmentation tool, it shifts the data synthesis from the feature space to a tree-induced decision space, ensuring that synthetic samples respect the underlying data distribution. It further reduces noisy synthesis through a consistency filtering step without incurring additional training cost. More importantly, TreeSMOTE readily combines with optimization-based approaches, representation learning, and undersampling ensembles. Providing diverse, structure-aware synthetic samples, it further improves established approaches and allows for more robust learning of imbalanced data.

# 3 Method

We first provide notations and preliminary knowledge of SMOTE and classification tree. Then, we define tree-induced structural similarity and gap based on the concept of lowest common ancestor and formally introduce the TreeSMOTE algorithm. In addition, since TreeSMOTE conducts data-level rebalancing and operates orthogonally to classification approaches, we show as an example how TreeSMOTE combines with undersampling ensembles, pushing the boundaries of their established state-of-the-art performance (Liu et al., 2025).

## 3.1 Preliminaries

### 3.1.1 Imbalanced Data Classification

We consider a classification problem on imbalanced tabular data $\mathcal{D} = \{(\boldsymbol{x}_i, y_i)\}_{i=1}^{N}$, where the feature $\boldsymbol{x}_i$ can be real and/or nominal valued, and the class label $y_i \in \{1, \ldots, C\}$. The dataset is imbalanced, with the majority and minority classes $\mathcal{D}_{\mathrm{maj}}$ and $\mathcal{D}_{\mathrm{min}}$ such that $\mathcal{D} = \mathcal{D}_{\mathrm{maj}} \cup \mathcal{D}_{\mathrm{min}}$ and $|\mathcal{D}_{\mathrm{maj}}| \gg |\mathcal{D}_{\mathrm{min}}|$, where $|.|$ denotes the cardinality. The class imbalance ratio (IR) is defined as $\mathrm{IR} = |\mathcal{D}_{\mathrm{maj}}|/|\mathcal{D}_{\mathrm{min}}|$. A higher IR indicates a more severe imbalance, which often biases classifiers toward the majority class.

### 3.1.2 SMOTE Overview

The synthetic minority oversampling technique (SMOTE) is a widely used to generate synthetic samples, often for minority classes, via interpolation between existing instances. Given a sample $\boldsymbol{x}_i$ and one of its spatially nearest neighbors $\boldsymbol{x}_j$ in the same class, a new sample is generated as

$$\boldsymbol{x}_i^{\text{syn}} = \boldsymbol{x}_i + \lambda(\boldsymbol{x}_j - \boldsymbol{x}_i), \quad \lambda \sim \text{Uniform}(0, 1). \tag{1}$$

Extant SMOTE approaches largely encode nominal features into one-hot vectors and apply Equation (1) without distinguishing from real-valued features. SMOTE increases the number of minority samples and mitigates class imbalance, but may generate misleading samples, particularly when local smoothness does not hold, and thus interpolations do not align with the underlying data distribution (e.g., Figure 1).

### 3.1.3 Trees and Hierarchical Structures

A classification and regression tree predicts a target label by learning decision rules inferred from features. The model is structured as a binary tree, where each internal node represents a question about a specific input feature, and each branch represents the outcome of that test. The process begins at the root node, where the algorithm selects the feature and specific split point that most effectively partitions the data into more homogeneously labeled subsets. For regression trees where the label is continuous, the purity of a node is usually based on minimizing the mean squared error. Tree models are more commonly used for classification problems, where the purity can be measured by the Gini index or information gain (Loh, 2011).

The recursive partitioning continues down the tree until a stopping criterion is met, such as a maximum depth or a minimum number of samples, at which point the final leaf nodes provide the predicted continuous or categorical label. Features chosen near the root typically have higher importance for classification, whereas deeper splits often involve less discriminative features. Each sample follows a path from the root to a leaf determined by these hierarchical decisions. For a given sample $\boldsymbol{x}_i$, its decision path is $\pi(\boldsymbol{x}_i) = \big(v_{i,1}, v_{i,2}, \ldots, v_{i,L_i}\big)$, where $v_{i,k}$ denotes the $k$-th visited node, and $L_i$ is the depth of the leaf reached by $\boldsymbol{x}_i$.

### 3.2 TreeSMOTE for Structure-Aware Data Augmentation

### 3.2.1 LCA Depth and Similarity

To quantify structural similarity with respect to feature importance and dependence, we leverage *lowest common ancestor* (LCA), which can be defined in tree models as the deepest ancestor of two nodes (Harel & Tarjan, 1984; Bender & Farach-Colton, 2000). We define the *LCA depth* for two samples $\boldsymbol{x}_i$ and $\boldsymbol{x}_j$ along their decision paths in a tree as

$$\text{Depth}_{\text{LCA}}(\boldsymbol{x}_i, \boldsymbol{x}_j) = \max\big\{\ell : v_{i,k} = v_{j,k}, \ \forall k = 1, \ldots, \ell\big\}, \tag{2}$$

where $v_{i,k}$ and $v_{j,k}$ denote the $k$-th nodes visited by $\boldsymbol{x}_i$ and $\boldsymbol{x}_j$ along their respective paths $\pi(\boldsymbol{x}_i)$ and $\pi(\boldsymbol{x}_j)$ from the root.

A larger $\text{Depth}_{\text{LCA}}(\boldsymbol{x}_i, \boldsymbol{x}_j)$ indicates that the two samples share a longer decision path along features starting from the root, which are often of higher importance in label prediction than those far from the root, and they remain indistinguishable by earlier splits of the tree. Therefore, $\text{Depth}_{\text{LCA}}$ provides a structure-aware measure of similarity in the partitioned feature space (or equivalently, the decision space) induced by the tree, and a larger value indicates a higher degree of similarity.

While a tree defines a hierarchical partition of the feature space, it is prone to overfit, sensitive to specific features, and of high variance (Loh, 2011). To addess the limitations, we consider multiple trees, $\{Tree^{(t)}\}_{t=1}^{T}$, where each tree is trained on a random subset of features, to facilitate the learning of various hierarchies of the feature space and provide a more robust and comprehensive measure of structural similarity. Note that this is different from random forests as we do not use boostrap samples. Specifically, we define the

tree-induced *structural similarity* between $\boldsymbol{x}_i$ and $\boldsymbol{x}_j$ across $T$ trees as

$$S(\boldsymbol{x}_i, \boldsymbol{x}_j) = \frac{1}{T} \sum_{t=1}^{T} \text{Depth}_{\text{LCA}}^{(t)}(\boldsymbol{x}_i, \boldsymbol{x}_j), \tag{3}$$

where $\text{Depth}_{\text{LCA}}^{(t)}(\boldsymbol{x}_i, \boldsymbol{x}_j)$ is calculated by Equation (2) based on tree $t$. Consequently, we define the tree-induced *structural gap* as

$$D(\boldsymbol{x}_i, \boldsymbol{x}_j) = H - S(\boldsymbol{x}_i, \boldsymbol{x}_j), \tag{4}$$

where $H$ is the maximum path length (i.e., the maximum tree depth). A larger gap indicates that the two samples' decision paths diverge earlier. The structural gap in Equation (4) that integrates information from multiple trees comprehensively captures feature hierarchies and dependence, providing a more reliable measure of structural proximity between samples.

In Appendix A, we provide a proposition establishing the ultrametric property of the structural gap in a one-tree configuration. This property is absent among spatial proximity metrics, like Euclidean distance, distinguishing the tree-induced structural similarity from spatial proximity. Instances with a short Euclidean distance are geometrically close but can diverge early along a decision path. In contrast, a smaller structural gap indicates that the two instances share longer decision paths across the $T$ trees. Since a tree splits nodes by reducing node impurity, leading to increasingly homogeneous label distributions in deeper nodes, two instances with a shorter structural gap are within a more refined and label-homogeneous region where local smoothness is better preserved.

### 3.2.2 Structure-Aware Data Synthesis

We formally introduce synthetic data generation by TreeSMOTE. For each (minority) instance $\boldsymbol{x}_i$, we first select at random one of its $k$ structurally similar samples of the same class, where the distance is measured by $D(\boldsymbol{x}_i, .)$ Then, synthetic samples can be generated by Equation (1) using the sample $x_i$ and one of its structurally similar samples $x_j$. The data generation process is repeated until a desired number of synthetic samples is obtained.

Nearest neighbors quantified by the tree-induced structural gap tend to be in a label-homogeneous region, where local smoothness of data space is better preserved. Restricting the synthesis using structurally similar samples, TreeSMOTE reduces the risk of generating noisy instances and preserves the intrinsic interdependence among features. By generating synthetic samples from structurally nearest neighbors, TreeSMOTE promotes label consistency and reduces label noise. Our empirical evaluation shows that a one-tree configuration of TreeSMOTE (i.e., $T = 1$) can already outperform extant SMOTE variants. Notably, performance gains scale positively with the number of trees until $T$ reaches the range of 50 to 100.

### 3.2.3 Consistency Filtering

To further improve the reliability of the data augmentation, each synthetic instance $\boldsymbol{x}_i$ is evaluated by $\{Tree^{(t)}\}_{t=1}^{T}$ without re-training:

$$\hat{y}_i^{\text{syn}} = \text{mode}\Big(\{\hat{y}^{(t)}(\boldsymbol{x}_i^{\text{syn}})\}_{t=1}^{T}\Big), \tag{5}$$

where $\hat{y}^{(t)}(\boldsymbol{x}_i^{\text{syn}})$ is the predicted label for $\boldsymbol{x}_i^{\text{syn}}$ by $Tree^{(t)}$, and mode$(\cdot)$ returns the most frequently predicted class among all the $T$ trees (i.e., majority voting). The synthetic instance is retained only if $\hat{y}_i^{\text{syn}} = y_i$, that is, its predicted label matches the intended class. This step eliminates instances that cross the decision boundaries induced by the $T$ trees, further mitigating noisy data generation. Note that SMOTE-TomekLinks and SMOTE-ENN use Tomek Links or an edited nearest neighbor rule to remove ambiguous synthetic data (Batista et al., 2004) but incur extra computational burden. In contrast, the consistency filtering step of TreeSMOTE utilizes the trained trees and requires negligible testing overhead.

To wrap up, TreeSMOTE consists of a data generation step and a consistency filtering step. The data generation step uses samples with a short structural gap to general structurally proximal synthetic data. The

---

**Algorithm 1** TreeSMOTE for Minority Augmentation

1: **Input:** $\mathcal{D} = \mathcal{D}_{maj} \cup \mathcal{D}_{min}$, $k$, $T$, $N_{gen}$
2: **Output:** $\tilde{\mathcal{D}}_{min}$
3: $\tilde{\mathcal{D}}_{min} \leftarrow$ null set
4: Train $\{\text{Tree}^{(t)}\}_{t=1}^{T}$ on $\mathcal{D}$
5: **while** $|\tilde{\mathcal{D}}_{min}| < N_{gen}$ **do**
6:     Sample $\boldsymbol{x}_i \in \mathcal{D}_{min}$
7:     Find $\boldsymbol{x}_j$ as one of $k$ structurally nearest neighbor of the same class, where Eq. (4) is used as to calculate proximity
8:     Generate $\boldsymbol{x}_i^{\text{syn}}$ via Eq. (1)
9:     Find $\hat{y}_i^{\text{syn}}$ by Eq. (5)
10:    **if** $\hat{y}_i^{\text{syn}} = y_i$ **then**
11:        Add $\{\boldsymbol{x}_i^{\text{syn}}, y_i\}$ to $\tilde{\mathcal{D}}_{min}$
12:    **end if**
13: **end while**
14: **return** $\tilde{\mathcal{D}}_{min}$

---

**Algorithm 2** TreeSMOTE-Augmented Ensemble

1: **Input:** $\mathcal{D} = \mathcal{D}_{maj} \cup \mathcal{D}_{min}$, $M$, untrained base learners $\{f^{(m)}\}_{m=1}^{M}$
2: **Output:** Ensemble classifier $f(\cdot)$
3: **for** $m = 1$ to $M$ **do**
4:     Sample a subset $\mathcal{D}_{maj}^{(m)} \subset \mathcal{D}_{maj}$
5:     Construct a balanced set:

$$\mathcal{D}^{(m)} \leftarrow \mathcal{D}_{maj}^{(m)} \cup \mathcal{D}_{min}$$

6:     Use TreeSMOTE in Algorithm 1 to augment $\mathcal{D}^{(m)}$ for each class; the augmented data is denoted by $\tilde{D}^{(m)}$
7:     Train base learner $f^{(m)}$ on $\tilde{D}^{(m)}$
8: **end for**
9: Find the ensemble $f(\cdot)$ by Eq. (6)
10: **return** $f(\cdot)$

---

consistency filtering step eliminates misleading instances without introducing extra training costs, further reducing noise in the generated data. The implementation of TreeSMOTE is summarized as Algorithm 1.

### 3.3 Integrating TreeSMOTE with Undersampling Ensemble

TreeSMOTE as a data-level rebalancing framework is readily integrated with classificaiton approaches. We show as an example how TreeSMOTE combines with ensemble methods.

#### 3.3.1 Undersampling Ensembles

Undersampling ensemble methods are widely used for imbalanced classification. Instead of training a single classifier on the full imbalanced dataset, ensembles construct multiple balanced training subsets by iteratively and randomly reducing samples from the majority class. Specifically, in each iteration $m = 1, \ldots, M$, a subset of majority samples $\mathcal{D}_{\text{maj}}^{(m)} \subset \mathcal{D}_{\text{maj}}$ is randomly selected and combined with the minority samples to form a training subset $\mathcal{D}^{(m)} = \mathcal{D}_{\text{maj}}^{(m)} \cup \mathcal{D}_{\text{min}}$. To ensure balanced labels, the subset is constructed such that $|\mathcal{D}_{\text{maj}}^{(m)}| = |\mathcal{D}_{\text{min}}|$.

A base classifier $f^{(m)}$ is trained on each balanced subset $\mathcal{D}^{(m)}$, and the final prediction is obtained by aggregating the outputs of all the base classifiers:

$$f(x) = \text{Agg}\big(f^{(1)}(x), f^{(2)}(x), \ldots, f^{(M)}(x)\big), \tag{6}$$

where $\text{Agg}(\cdot)$ denotes an ensemble aggregation function such as majority voting or weighted averaging. Utilizing different majority subsets, the undersampling ensembles removes the dominance of the majority class and achieve diversity among base learners.

#### 3.3.2 TreeSMOTE-Augmented Ensemble

Undersampling ensembles rebalance data by using subsets of the majority and suffer from the limitations of data size and diversity. TreeSMOTE can be used in this context to address the limitations. For each undersampled training subset $\mathcal{D}^{(m)}$, synthetic data are generated for each class (majority and minority), and the TreeSMOTE-augmented subset of $\mathcal{D}^{(m)}$, denoted by $\tilde{\mathcal{D}}^{(m)}$, is used to train a base learner $f^{(m)}$. The final ensemble prediction is obtained by Equation (6), i.e., aggregation via majority voting or weighted averaging. We summarize this TreeSMOTE-augmented ensemble as Algorithm 2.

### 3.4 Related Work on Random Forest Proximities

In a random forest, proximity (also known as similarity) between two observations was originally defined as the proportion of trees in which they share the same leaf node (Cutler et al., 2012). Because task-relevant variables are more frequently selected for tree splits, this proximity inherently incorporates supervised variable importance. It has been used for a variety of tasks, such as clustering (Shi & Horvath, 2006), outlier detection (Nesa et al., 2018), dimensionality reduction, and data visualization (Pouyan et al., 2016; Rhodes et al., 2021). To mitigate overfit to training data, out-of-bag (OOB) proximity was introduced, which calculates the proportion using strictly the trees where the observations are OOB (Hastie et al., 2008). More recently, a proximity that integrates both in-bag and OOB observations has been developed to enable proximity-weighted predictions that effectively match the random forest's OOB predictions (Rhodes et al., 2023).

These forest proximities rely on leaf-node co-occurrence of two observations. However, observations that diverge at the very last split of a deep tree are similar, but these proximities treat them as completely disjoint. Relaxing the leaf-node co-occurrence constraint brings attention to these near-matches and can increase robustness against noisy splits near the leave nodes. Our proposed tree-induced structural similarity relaxes this constraint by quantifying proximity based on the deepest ancestor of two observations. Even if two observations do not reside in the same leaf node, they can remain closely aligned along a decision path. So, the structural similarity allows interpolation between observations in a continuous way. A related random forest proximity relaxing the leaf-node co-occurrence constraint accounts for the number of branches between observations within a tree (Englund & Verikas, 2012), but it has been specifically designed to construct the kernel for support vector machines rather than for data augmentation.

## 4 Experiments and Results

We show consistent improvement with TreeSMOTE by comparing it against SMOTE-based and generative-model-based augmentation techniques. Furthermore, we demonstrate TreeSMOTE's ability to combine with undersampling ensembles, extending the capabilities of current state-of-the-art approaches for imbalanced tabular learning. Our evaluation spans 83 datasets with diverse imbalance levels, ranging from nearly balanced to extremely skewed with imbalance ratios over 100. In addition, we evaluate TreeSMOTE by using it with various classification models. The competitive performance of TreeSMOTE under a default setting (see Appendix B) demonstrates its effectiveness as an out-of-the-box tool for imbalanced tabular classification.

### 4.1 Experiment Setup

We evaluate the proposed method on the CLIMB benchmark (Liu et al., 2025), a large-scale collection of 73 datasets with varying levels of class imbalance. The imbalance ratios (IRs) are categorized as low (IR $< 5$), medium ($5 \leq$ IR $< 10$), high ($10 \leq$ IR $< 50$), and extreme (IR $\geq 50$). The data we use have sizes varying from a few hundred to several hundred thousand and include 60 binary-class datasets and 13 multi-class datasets. Since the CLIMB benchmark is primarily evaluated using classical imbalance-learning methods and its large scale poses computational challenges for neural network training, we additionally include 10 datasets to facilitate comparisons with generative-model-based approaches. These datasets are adopted from generative-model-based imbalanced learning literature (Ren et al., 2025) and are available in the Python package *imbalanced-learn* (Lemaître et al., 2017). We provide detailed information about datasets in Table 6 and Table 7 in Appendix B.1.

To thoroughly evaluate performance, especially on minority classes, we follow Liu et al. (2025; 2020) to assess the methods using multiple metrics that capture different aspects of performance, including the Area Under the Precision–Recall Curve (AUPRC) (Davis & Goadrich, 2006), F1-score (Sokolova et al., 2006), Matthews Correlation Coefficient (MCC) (Boughorbel et al., 2017), and the geometric mean of precision and recall(G-mean) (Powers, 2020). The detailed definitions of these metrics are provided in Appendix B.2.

We compare the proposed TreeSMOTE with a diverse set of baselines. For classical oversampling methods, we compare with widely used SMOTE-based approaches, including SMOTE (Chawla et al., 2002), BorderlineSMOTE (Han et al., 2005), SVMSMOTE (Nguyen et al., 2011), and SMOTE-TomekLinks (Batista et al., 2004). For generative-model-based methods, we consider TVAE, CTGAN (Xu et al., 2019), CTGAN-ENN (Adiputra & Wanchai, 2024), and two diffusion models, DGOT (Ren et al., 2025) and TabDDPM (Kotelnikov et al., 2023), which are modern generative models specifically designed for imbalanced tabular learning. For each dataset and method, we use five classifiers: Classification Tree, LightGBM (Ke et al., 2017), multi-layer perceptron (MLP), LinearSVC, and TabM (Gorishniy et al., 2025). In addition, when integrating our method with ensemble-based approaches, we evaluate its performance with several ensemble methods that have reported state-of-the-art performance for imbalanced tabular learning (Liu et al., 2025), including Self-paced Ensemble (Liu et al., 2020), Balanced Random Forest (Khoshgoftaar et al., 2007), RUSBoost (Seiffert et al., 2009), UnderBagging (Barandela et al., 2003), and BalanceCascade (Liu et al., 2008). Notably, the competitive performance of TreeSMOTE is further validated by its comparable performance to Self-paced Ensemble. We defer detailed model implementations to Appendix B.

## 4.2 Results

In the following tables, the best and second-best results are indicated by bold and underlined numbers, respectively. Performance metrics for the proposed TreeSMOTE are highlighted with gray shading. Results represent the mean performance $\pm$ standard error across five training-testing partitions. We defer more results to the appendix. The comparison with baselines under default configurations and Self-paced Ensemble, which is among the most advanced imbalanced tabular classification methods, underpins TreeSMOTE as an effective out-of-the-box data augmentation tool for imbalanced tabular classification.

### 4.2.1 SMOTE-Based Methods

We first compare our method with SMOTE-based approaches on the CLIMB benchmark, which contains 73 datasets grouped into four imbalance levels: low (28), medium (24), high (15), and extreme (6). Table 1 reports the average performance using AUPRC, BAC, F1-Score, G-mean, and MCC. Detailed results for each respective evaluation metric are provided in Appendix C.1.

The results show that TreeSMOTE achieves consistent improvements across all imbalance levels and all four downstream classifiers. It delivers the best results in most settings. In particular, under highly imbalanced settings, the performance of most extant SMOTE-based methods tends to degrade significantly. This is especially prominent when using simpler downstream models such as Classification Tree and LinearSVC, where synthetic samples generated by conventional SMOTE-based methods may introduce noisy or ambiguous instances that distort the decision boundary. In contrast, TreeSMOTE is able to preserve and even improve classification performance under these challenging scenarios. To further illustrate the effectiveness of TreeSMOTE compared to other SMOTE-based methods, Table 2 reports the number of the 73 CLIMB datasets on which each method achieves either the best or second-best performance for each evaluation metric, averaged across downstream classifiers. These results indicate that TreeSMOTE delivers an overall better performance across diverse datasets and metrics.

When using a downstream classifier with higher capacity, such as MLP, TabM, and LightGBM, the baseline performance is considerably stronger than that of classification tree and LinearSVC. While other SMOTE-based methods can provide performance improvement in this scenario, TreeSMOTE improves the most and achieves overall better results, indicating that the samples generated by our method better capture the underlying data structure and benefit downstream learning. Moreover, the results on the 73 CLIMB datasets (and the 10 *imbalanced-learn* datasets, shortly) suggest using LightGBM as the downstream classifier for TreeSMOTE generally delivers better performance.

### 4.2.2 Generative-Model-Based Methods

For a more comprehensive comparison, we evaluate TreeSMOTE against four popular generative models for tabular data generation: TVAE, CTGAN, CTGAN-ENN, DGOT, and TabDDPM. We report the average performance across the five evaluation metrics using five downstream classifiers (Classification Tree, Lin-

Table 1: Performance comparison (mean ± standard error) among SMOTE-based minority data augmentation. Reported values are averaged over AUPRC, F1-Score, Balanced Accuracy, MCC, and G-mean. The best and second-best results are highlighted in bold and underlined, respectively.

| Classifier | Oversampling Method | Imbalance Level | | | | |
|---|---|---|---|---|---|---|
| | | Low | Medium | High | Extreme | Average |
| Classification Tree | None | 61.82±0.09 | 65.99±0.09 | 51.67±0.11 | 65.02±0.31 | 61.37±0.06 |
| | SMOTE | 62.54±0.09 | 65.90±0.08 | 51.88±0.11 | 60.07±0.32 | 61.25±0.06 |
| | BorderlineSMOTE | 61.82±0.10 | 65.97±0.09 | 51.59±0.09 | 61.31±0.28 | 61.04±0.06 |
| | SVMSMOTE | 62.13±0.10 | 66.36±0.10 | 52.23±0.09 | 62.30±0.21 | 61.50±0.06 |
| | SMOTE-TomekLinks | **62.80±0.10** | 66.37±0.09 | 51.42±0.11 | 60.74±0.20 | 61.46±0.06 |
| | TreeSMOTE | 62.73±0.10 | **66.83±0.09** | **52.35±0.08** | **65.17±0.32** | **62.15±0.05** |
| LinearSVC | None | 66.31±0.09 | 67.07±0.07 | 46.61±0.08 | 59.25±0.13 | 61.93±0.04 |
| | SMOTE | 68.46±0.10 | 69.72±0.07 | 50.15±0.08 | 57.59±0.15 | 64.22±0.05 |
| | BorderlineSMOTE | 67.60±0.10 | 68.40±0.07 | 49.10±0.08 | 58.84±0.18 | 63.34±0.05 |
| | SVMSMOTE | 67.98±0.10 | 69.38±0.07 | 49.83±0.10 | 60.63±0.19 | 64.11±0.05 |
| | SMOTE-TomekLinks | 68.57±0.10 | 68.89±0.07 | 48.79±0.07 | 56.13±0.19 | 63.59±0.05 |
| | TreeSMOTE | **69.10±0.10** | **70.46±0.07** | **50.76±0.09** | **62.82±0.19** | **65.26±0.05** |
| MLP | None | 69.96±0.08 | 72.41±0.07 | 57.69±0.11 | 66.49±0.26 | 67.96±0.05 |
| | SMOTE | 70.97±0.09 | 72.91±0.07 | 58.47±0.10 | 68.68±0.20 | 68.85±0.05 |
| | BorderlineSMOTE | 70.18±0.08 | 72.81±0.07 | 58.75±0.10 | 69.21±0.24 | 68.62±0.05 |
| | SVMSMOTE | 70.64±0.09 | 73.32±0.07 | 59.41±0.10 | 69.09±0.26 | 69.08±0.05 |
| | SMOTE-TomekLinks | 70.99±0.10 | 72.50±0.07 | 57.78±0.10 | 67.95±0.21 | 68.52±0.05 |
| | TreeSMOTE | **71.37±0.08** | **73.42±0.07** | **59.42±0.11** | **70.28±0.25** | **69.50±0.05** |
| TabM | None | 66.14±0.13 | 71.17±0.07 | 57.77±0.10 | 60.02±0.19 | 65.68±0.06 |
| | SMOTE | 68.19±0.11 | 72.00±0.08 | 60.70±0.12 | 66.09±0.34 | 67.83±0.06 |
| | BorderlineSMOTE | 68.03±0.09 | 72.05±0.08 | 61.51±0.13 | 66.55±0.33 | 67.98±0.06 |
| | SVMSMOTE | 68.21±0.08 | 72.70±0.08 | 61.46±0.11 | 67.12±0.33 | 68.30±0.05 |
| | SMOTE-TomekLinks | 68.25±0.10 | 71.99±0.08 | 60.08±0.11 | 65.10±0.30 | 67.65±0.06 |
| | TreeSMOTE | **68.44±0.09** | **72.94±0.07** | **62.04±0.10** | **68.31±0.35** | **68.69±0.05** |
| LightGBM | None | 70.63±0.08 | 74.03±0.08 | 61.84±0.14 | 64.30±0.41 | 69.53±0.06 |
| | SMOTE | 71.46±0.09 | 74.86±0.06 | 63.40±0.11 | 71.64±0.23 | 71.04±0.05 |
| | BorderlineSMOTE | 71.44±0.09 | 74.64±0.06 | 63.65±0.10 | 70.97±0.36 | 70.95±0.05 |
| | SVMSMOTE | 71.43±0.09 | 75.13±0.06 | 63.70±0.11 | 71.53±0.28 | 71.17±0.05 |
| | SMOTE-TomekLinks | **71.62±0.09** | 74.97±0.08 | 62.88±0.10 | 68.36±0.29 | 70.77±0.05 |
| | TreeSMOTE | 71.26±0.09 | **75.02±0.07** | **64.01±0.10** | **73.85±0.30** | **71.32±0.05** |

Table 2: Number of CLIMB datasets on which a SMOTE-based method performs at least the second best (best + second best) across different metrics.

| Method | AUPRC | BAC | F1-score | G-mean | MCC | Overall |
|---|---|---|---|---|---|---|
| None | 38 (29+9) | 3 (2+1) | 24 (14+10) | 26 (14+12) | 33 (20+13) | 124 (79+45) |
| SMOTE | 10 (7+3) | 32 (12+20) | 12 (3+9) | 11 (3+8) | 10 (4+6) | 75 (29+46) |
| BorderlineSMOTE | 11 (5+6) | 16 (8+8) | 10 (4+6) | 12 (4+8) | 14 (4+10) | 63 (25+38) |
| SVMSMOTE | 19 (8+11) | 35 (24+11) | 31 (8+23) | 29 (10+19) | 21 (8+13) | 135 (58+77) |
| SMOTE-TomekLinks | 14 (8+6) | 32 (13+19) | 11 (6+5) | 12 (7+5) | 19 (8+11) | 88 (42+46) |
| TreeSMOTE | 54 (16+38) | 28 (14+14) | 58 (38+20) | 56 (35+21) | 49 (29+20) | 245 (132+113) |

Table 3: Performance comparison (mean ± standard error) among minority data augmentation by a generative model or TreeSMOTE on 10 *imbalanced-learn* datasets. Reported values are averaged over five metrics (AUPRC, F1-Score, Balanced Accuracy, MCC, and G-mean) and five downstream classifiers (Classification Tree, LinearSVC, MLP, LightGBM, and TabM). The best results are highlighted in bold, and the second-best results are underlined.

| Dataset | TVAE | CTGAN | CTGAN-ENN | DGOT | TabDDPM | TreeSMOTE |
|---|---|---|---|---|---|---|
| optical_digits | 88.21±0.07 | 85.00±0.07 | 91.19±0.10 | 92.02±0.08 | 92.03±0.05 | **92.95±0.08** |
| satimage | 58.11±0.13 | 55.83±0.09 | 52.39±0.15 | 55.76±0.19 | 58.66±0.15 | **61.69±0.15** |
| pen_digits | 92.14±0.05 | 91.91±0.02 | 91.18±0.07 | 94.75±0.08 | 93.28±0.03 | **95.19±0.05** |
| sick_euthyroid | 77.26±0.31 | 67.99±0.25 | 64.47±0.68 | 83.00±0.17 | 82.51±0.13 | **84.30±0.16** |
| thyroid_sick | 73.08±0.27 | 66.24±0.30 | 62.13±0.48 | 78.19±0.37 | 80.55±0.15 | **83.28±0.09** |
| coil_2000 | 35.92±0.08 | 34.26±0.14 | 32.92±0.05 | 34.13±0.03 | 35.76±0.04 | **37.32±0.05** |
| wine_quality | 50.07±0.20 | 45.51±0.25 | 43.00±0.19 | 42.14±0.20 | 46.84±0.18 | **51.55±0.29** |
| letter_img | 86.35±0.17 | 86.93±0.13 | 88.56±0.06 | **93.29±0.05** | 89.93±0.06 | 92.71±0.05 |
| abalone_19 | 31.53±0.19 | **32.25±0.17** | 31.20±0.06 | 30.39±0.03 | 31.20±0.09 | 31.92±0.26 |
| isolet | 85.57±0.06 | 86.47±0.05 | 82.94±0.04 | 85.48±0.06 | 86.82±0.03 | **87.75±0.05** |

Table 4: Number of *imbalanced-learn* datasets on which a tabular generative model performs at least the second best (best + second best) across different metrics.

| Method | AUPRC | BAC | F1-score | G-mean | MCC | Overall |
|---|---|---|---|---|---|---|
| TVAE | 1 (0+1) | 4 (1+3) | 3 (0+3) | 2 (0+2) | 2 (0+2) | 12 (1+11) |
| CTGAN | 1 (0+1) | 3 (1+2) | 0 (0+0) | 1 (0+1) | 1 (1+0) | 6 (2+4) |
| CTGAN-ENN | 0 (0+0) | 0 (0+0) | 0 (0+0) | 0 (0+0) | 0 (0+0) | 0 (0+0) |
| DGOT | 7 (3+4) | 0 (0+0) | 4 (2+2) | 4 (2+2) | 4 (1+3) | 19 (8+11) |
| TabDDPM | 1 (0+1) | 4 (0+4) | 3 (0+3) | 3 (0+3) | 3 (0+3) | 14 (0+14) |
| TreeSMOTE | 10 (7+3) | 9 (8+1) | 10 (8+2) | 10 (8+2) | 10 (8+2) | 49 (39+10) |

earSVC, MLP, LightGBM, and TabM) on 10 datasets from the Python package *imbalanced-learn* (Lemaître et al., 2017). The detailed results are reported in Table 3. The results show that TreeSMOTE achieves at least the second-best performance in all the settings and the best performance in most of the settings, demonstrating its excellent and consistent performance across datasets and downstream classifiers.

Among the baseline methods, DGOT and TabDDPM perform better than other generative approaches but underperforms TreeSMOTE, though it is specifically designed for oversampling and imbalanced tabular data learning. TVAE and CTGAN achieve moderate performance overall, possibly because they are designed as general-purpose tabular data generators. Furthermore, CTGAN-ENN performs poorly in most settings. One possible reason is that this method is originally designed for applications that may differ from the benchmark data in this paper, limiting its generalizability when applied to diverse imbalanced classification datasets. Detailed results for other metrics are provided in Appendix C.2. Table 4 reports the number of the *imbalanced-learn* datasets for which each method achieves the best or second-best performance across five evaluation metrics, averaged over the downstream classifiers. Notably, TreeSMOTE attains top-two performance on all 10 datasets evaluated by AUPRC, F1-score, G-mean, and MCC and 9 datasets evaluated by BAC.

### 4.2.3 Ensemble-Based Methods

A data augmentation approach, especially a generative model, can be sensitive to its hyperparameters. In practice, exhaustively tuning hyperparameters to maximize a method's performance can be computationally prohibitive for end users, entailing good performance under default settings. This constraint applies directly to our study, given the large number of datasets and baseline methods evaluated. To assess the out-of-the-box performance of TreeSMOTE, we compare it with the Self-paced Ensemble, one of the top-performing approaches for imbalanced tabular classification (Liu et al., 2025) that exhibits high robustness to hyperparameter variations within a reasonable range (Liu et al., 2020). Figure 2 illustrates the performance of TreeSMOTE relative to the Self-paced Ensemble, both under the default configurations. Both methods use LightGBM as the downstream classifier as they yield better results than using the other four evaluated classifiers. Figure 2 (a) illustrates the results for the 73 CLIMB datasets, and Figure 2 (b) covers the 10 *imbalanced-learn* datasets. The baseline performance of LightGBM on the original data is also included. Notably, TreeSMOTE performs comparably to or slightly outperforms the Self-paced Ensemble across all metrics except for BAC, demonstrating its strong out-of-the-box efficacy.

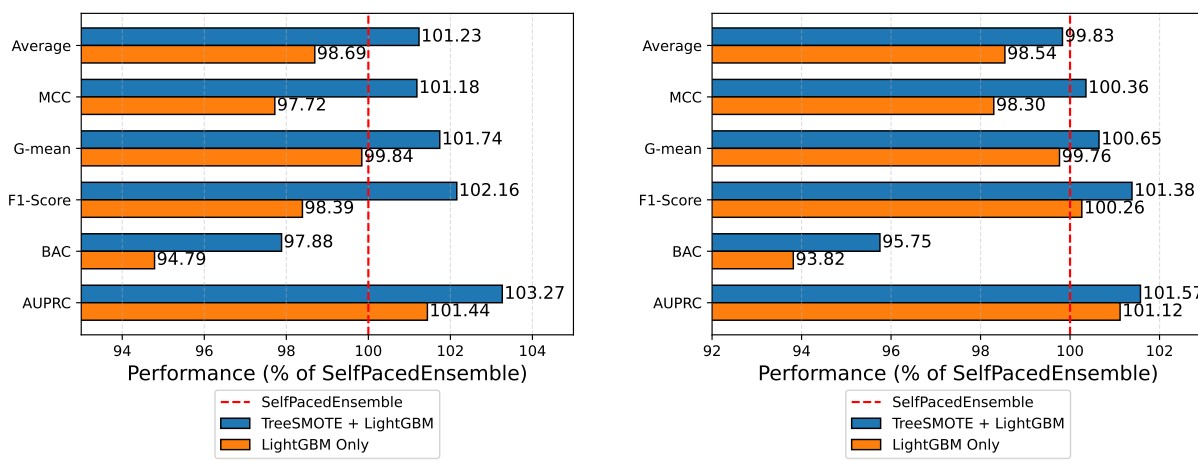

(a) Average performance on the 73 CLIMB datasets.      (b) Average performance on the 10 *imbalanced-learn* datasets.

Figure 2: Performance of TreeSMOTE relative to Self-paced Ensemble. Both use lightGBM as the downstream classifier. Values over and under 100% indicates better and worse performance, respectively.

We further combine TreeSMOTE with several undersampling ensemble approaches, including SelfPacedEnsemble, BalancedRandomForest, RUSBoost, UnderBagging, and BalanceCascade, which are among the most advanced methods for imbalanced tabular classification. Experiments are conducted on the CLIMB datasets. Tables 5 report the results in terms of the average over the five evaluation metrics. As a result, our method consistently improves the performance of these competitive ensemble approaches in most settings. In addition, the improvement becomes more pronounced as the imbalance ratio increases. This can be at-

Table 5: Performance comparison averaging over AUPRC, BAC, F1-Score, G-mean, and MCC between ensemble methods and their combinations with TreeSMOTE. Reported values are averaged over AUPRC, F1-Score, Balanced Accuracy, MCC, and G-mean. For each pair, the better result is highlighted in bold.

| Method | Low IRs | Medium IRs | High IRs | Extreme IRs | Average |
|---|---|---|---|---|---|
| SelfPacedEnsemble | **71.38±0.09** | 75.08±0.06 | 60.98±0.08 | 71.73±0.20 | 70.45±0.05 |
| SelfPacedEnsemble w/ TreeSmote | 71.28±0.10 | **75.32±0.06** | **61.21±0.07** | **71.97±0.22** | **70.57±0.05** |
| BalancedRandomForest | **71.15±0.10** | 73.48±0.06 | 57.45±0.07 | 59.01±0.16 | 67.97±0.04 |
| BalancedRandomForest w/ TreeSmote | 70.94±0.09 | **73.65±0.06** | **57.88±0.07** | **59.23±0.16** | **68.07±0.04** |
| RUSBoost | 70.46±0.11 | 72.84±0.06 | 57.02±0.08 | **61.68±0.22** | 67.64±0.05 |
| RUSBoost w/ TreeSmote | **70.72±0.10** | **73.45±0.07** | **57.66±0.09** | 61.56±0.21 | **68.08±0.05** |
| UnderBagging | **71.09±0.10** | 74.16±0.07 | 58.98±0.06 | 60.75±0.18 | 68.66±0.05 |
| UnderBagging w/ TreeSmote | 71.05±0.10 | **74.17±0.06** | **59.24±0.08** | **61.01±0.15** | **68.73±0.05** |
| BalanceCascade | 70.57±0.09 | **74.29±0.06** | 56.30±0.09 | 67.57±0.24 | 68.53±0.05 |
| BalanceCascade w/ TreeSmote | **71.02±0.10** | 74.02±0.07 | **56.30±0.08** | **68.02±0.20** | **68.64±0.05** |

tributed to the fact that TreeSMOTE has diversified the sample distribution of each subset constructed by the undersampling ensemble and enhanced the representation of both minority and majority classes. By generating additional samples guided by tree-based partitions, TreeSMOTE helps alleviate the information loss introduced by undersampling while preserving the local decision structure, leading to improved classification performance. Other detailed results are provided in Appendix C.3.

## 5 Sensitivity and Additional Analysis

### 5.1 Impact of the Number of Trees

Figure 3 presents the average F1-score, G-mean, and MCC of TreeSMOTE combined with an MLP classifier across all datasets in the CLIMB benchmark, as a function of the number of trees. Even with one tree, TreeSMOTE outperforms other SMOTE variants across all three metrics. Performance improves consistently as the number of trees increases, with gains diminishing after approximately 50 trees. Despite insignificance, the slight performance decrease over 50 trees aligns with the finding of non-monotonic performance of a random forest with the number of trees (Probst & Boulesteix, 2018). The result indicates that while a reasonably large number of trees can be beneficial, a small to moderate ensemble suffices for a strong performance of TreeSMOTE.

### 5.2 Noisy Features or Labels

We evaluate the robustness of TreeSMOTE on the 10 *imbalanced-learn* datasets in the presence of noisy features or labels. To introduce feature noise, we randomly select a subset of training data and permute half of the feature values within each selected sample. This within-sample feature permutation substantially disrupts the tree decision boundaries. To introduce label noise, we randomly flip the labels of selected samples to a different class.

We vary the proportion of samples with noisy features from 10% to 50%. The proportion of noisy labels ranges from 10% to 30%, and we do not consider a higher proportion because it reduces class imbalance. We use tree, LinearSVC, MLP and LightGBM as downstream classifiers and report the results in Figure 4. In the presence of noisy features (Figure 4, row 1), the TreeSMOTE's performance degradation is moderate as the noise level increases, and it is comparable to SMOTE and better than without oversampling. However, in the presence of noisy labels (Figure 4, row 2), both TreeSMOTE and SMOTE suffer considerable performance degradation, yielding results comparable to the baseline without data augmentation. This result suggests an

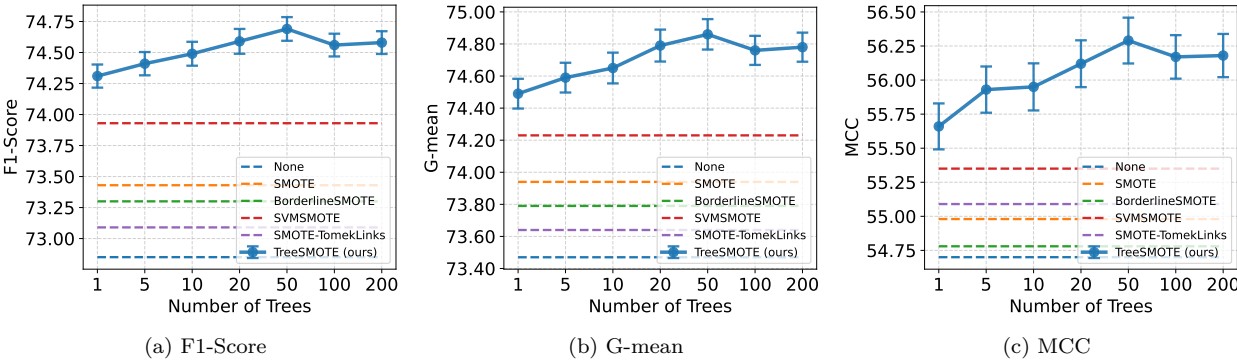

(a) F1-Score      (b) G-mean      (c) MCC

Figure 3: Average F1-score, G-mean, and MCC across all datasets with respect to the number of trees, using MLP as the downstream classifier.

application boundary of TreeSMOTE; high levels of data noise undermine the reliability of the tree decision rules and the structural similarity between minority samples.

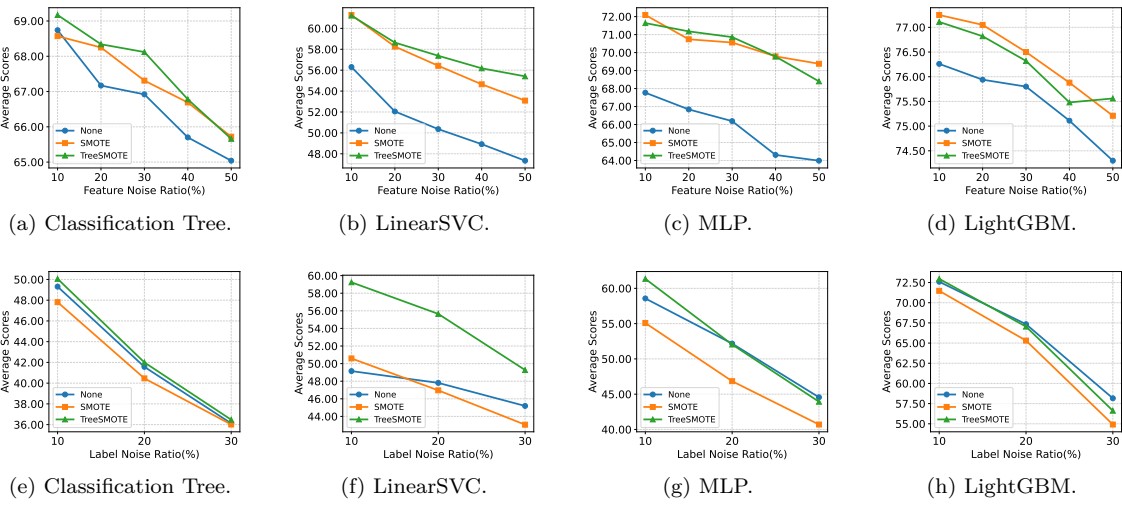

(a) Classification Tree.    (b) LinearSVC.    (c) MLP.    (d) LightGBM.

(e) Classification Tree.    (f) LinearSVC.    (g) MLP.    (h) LightGBM.

Figure 4: Performance evaluation under varying feature and label noise levels. Row 1: Feature noise is introduced by randomly permuting 50% of feature values within a selected sample. The proportion of corrupted training samples varies from 10% to 50%. Row 2: Label noise is introduced by randomly flipping class labels. The proportion of corrupted training samples varies from 10% to 30%. The reported score is averaged over AUPRC, F1-Score, Balanced Accuracy, MCC, and G-mean.

### 5.3 Additional Results

We provide additional experimental results in the appendix. Appendix C.6 presents t-SNE visualizations of the data distributions before and after oversampling on two real-world datasets (*mfeat-zernike* and *Pulsar-Dataset-HTRU2*). Compared with SMOTE, TreeSMOTE generates synthetic samples that better preserve class boundaries and introduce less noise (see Figure. 5 in the appendix). Appendix C.7 reports an ablation study evaluating two key components of TreeSMOTE: the tree-induced structural gap versus Euclidean distance, and the consistency filtering step. The results in Table 27 in the appendix demonstrate that both the structural gap and consistency filtering contribute positively to overall performance, with the full configuration achieving the best results across most metrics. We compare the computational efficiency of data augmentation by the SMOTE-based methods and the tabular generative models. Table 28 in Appendix C.8

shows that TreeSMOTE requires longer time than SMOTE, BorderlineSMOTE, and SMOTE-TomekLinks because of its tree training and consistency filtering. When compared with generative models, TreeSMOTE demonstrates a significant computational advantage in terms of execution time and memory consumption without requiring GPU acceleration. The results establish TreeSMOTE as a practical out-of-the-box tool for tabular data augmentation.

## 6  Discussion and Conclusion

We introduce TreeSMOTE, a novel structure-aware oversampling framework designed to address the persistent challenges of class imbalance, particularly within tabular data. While extant SMOTE-based approaches rely on spatial proximity—often failing to capture complex data distributions or the dependencies of heterogeneous features—TreeSMOTE performs data augmentation within a tree-induced decision space. By utilizing the lowest common ancestor depth along decision paths to quantify structural similarity, our method ensures that synthetic samples are generated between structurally compatible instances, better preserving local smoothness and the intrinsic data manifold. Our t-SNE visualizations confirm that shifting to a hierarchical decision space produces high-quality instances that respect class boundaries, resulting in larger-margin class separation and a reduction in noisy artifacts.

Extensive empirical evaluations demonstrate that TreeSMOTE consistently delivers competitive performance across a wide range of imbalance levels, from nearly balanced to extremely skewed. It not only functions as an effective standalone data augmentation method but also seamlessly integrates with and improves classification models. In particular, under the default configuration, TreeSMOTE achieves comparable performance to Self-paced Ensembler, which is among the most advanced imbalanced tabular classification approaches. We also combine TreeSMOTE with undersampling ensembles and further improve their state-of-the-art performance. Overall, TreeSMOTE serves an out-of-the-box tool with no need for computationally expensive model selection, offering an effective and practical solution to end users for imbalanced tabular data classification.

**Limitation.**  Like many classification models, the trees in TreeSMOTE can be biased toward the majority, causing minority samples to be misclassified during oversampling and potentially undermining the robustness of the synthetic data. TreeSMOTE mitigates this issue through three mechanisms. First, the use of a random subset of features, as used in random forests, functionally acts as a dropout, helping regularize spurious features, which are a primary driver of bias in imbalanced datasets (Shi et al., 2024; Wen et al., 2025). Second, we interpolate between pairs of minority samples that share a longer decision path, regardless of whether they are correctly classified. Even if these samples are misclassified, this approach helps preserve the local smoothness between them. Third, the consistency filtering step eliminates synthetic instances with inconsistent predicted labels, further enhancing augmented data robustness. While we empirically observe that the consistency filtering improves overall performance, it may deliver lower synthetic data diversity compared to conventional SMOTE without a filtering step.

**Applicability Boundary.**  While the sensitivity analysis indicates that TreeSMOTE is reasonably robust to noisy features, it exhibits a vulnerability to noisy labels, where its performance degrades to levels comparable to using the original imbalanced data. One should be cautious when applying TreeSMOTE to domains with highly unreliable annotations, or consider employing label-denoising techniques before oversampling. Because TreeSMOTE relies on the integrity of the tree decisions to guide synthetic sample generation, corrupted labels can cause the method to interpolate between misleading regions, effectively amplifying the noise. Consequently, TreeSMOTE is best deployed in scenarios where label fidelity is reasonably assured, whereas heavily corrupted datasets necessitate prerequisite data sanitization. In addition, Tabular foundation models, such as TabPFN (Hollmann et al., 2022) and TabICL (Qu et al., 2025), have emerged as highly effective tools for tabular classification. We have evaluated the data augmentation methods with TabPFN-3 (Grinsztajn et al., 2026) as the downstream classifier and observed minimal performance improvement compared to training on the original, unaugmented data. This observation aligns with the findings of McDowell et al. (2026), who note minimal benefits of oversampling when using PFN as the classifier. So, this es-

tablishes another applicability boundary for TreeSMOTE: It offers little to no benefit when paired with advanced tabular foundation models that inherently achieve robust performance on imbalanced data.

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

## A  Ultrametric Property

**Proposition** (Ultrametric property of one-tree structural gap.)  Let $H^{(t)}$ be the depth of $Tree^{(t)}$. The tree-induced structural gap $D^{(t)}(\boldsymbol{x}_i, \boldsymbol{x}_j) = H^{(t)} - \text{Depth}_{\text{LCA}}{}^{(t)}(\boldsymbol{x}_i, \boldsymbol{x}_j)$ satisfies the ultrametric inequality $D^{(t)}(\boldsymbol{x}_i, \boldsymbol{x}_k) \leq \max\left(D^{(t)}(\boldsymbol{x}_i, \boldsymbol{x}_j), D^{(t)}(\boldsymbol{x}_j, \boldsymbol{x}_k)\right)$, for any $\boldsymbol{x}_i$, $\boldsymbol{x}_j$, and $\boldsymbol{x}_k$.

**Proof of Proposition A.** In a single tree, any three samples have a unique lowest common ancestor structure. Let $u = \text{LCA}(\boldsymbol{x}_i, \boldsymbol{x}_j)$ and $v = \text{LCA}(\boldsymbol{x}_j, \boldsymbol{x}_k)$. By the tree hierarchy, the LCA of $(\boldsymbol{x}_i, \boldsymbol{x}_k)$ occurs at least as high as the shallower of $u$ and $v$, which implies

$$\text{Depth}_{\text{LCA}}(\boldsymbol{x}_i, \boldsymbol{x}_k) \geq \min(\text{Depth}_{\text{LCA}}(\boldsymbol{x}_i, \boldsymbol{x}_j), \text{Depth}_{\text{LCA}}(\boldsymbol{x}_j, \boldsymbol{x}_k)).$$

This is equivalent to the proposition.

## B  Implementation Details

For SMOTE-based methods, the number of nearest neighbors is set to $k = 5$ for all datasets. For TVAE, CTGAN, CTGAN-ENN, and DGOT, we adopt the default hyperparameters provided in their official implementations (Xiao et al., 2021; Adiputra & Wanchai, 2024; Ren et al., 2025). For ensemble methods, we use the default hyperparameter settings provided by the benchmark library (Liu, 2025).

TreeSMOTE constructs a Random Forest with 100 trees to capture the hierarchical structure of the data. Each tree is grown using Gini impurity as the split criterion, with no depth restriction, and each leaf node contains at least one sample. At each split, a random subset of features is considered when searching for the best split, following the standard feature subsampling strategy used in Random Forests, where the number of candidate features is set to $\sqrt{d}$ for classification tasks, with $d$ denoting the total number of features. When combined with an undersampling ensemble method, TreeSMOTE performs class-wise data augmentation on each balanced subset, doubling the number of samples in each class.

Each experiment is repeated five times with different random seeds, and the average results are reported. For the downstream classifiers, Classification Tree, LinearSVC, and MLP use the default hyperparameters provided by the scikit-learn package, while LightGBM uses the default settings from its official implementation. For each dataset, samples are randomly split into 75% training and 25% testing before applying any resampling or augmentation. Nominal variables are encoded into one-hot vectors for fair comparison among different methods, as some of them are not capable of processing nominal variables without dummification.

### B.1  Summary statistics of datasets

We present summary statistics of all datasets used in our experiments, including the CLIMB benchmark (grouped by imbalance level) and the datasets from the imbalanced-learn package.

### B.2  Metrics

The evaluation metrics used in this study are defined as follows:

- Recall: $\frac{TP}{TP+FN}$, where $TP$ is the number of true positives and $FN$ is the number of false negatives. It measures the ability of the model to correctly identify positive instances.

- Precision: $\frac{TP}{TP+FP}$, where $FP$ is the number of false positives. It measures the accuracy of the positive predictions made by the model.

- AUPRC: The area under the precision-recall curve, which emphasizes the precision-recall trade-off for the positive (typically minority) class, making it particularly informative in imbalanced settings.

- BAC: $\frac{1}{2}\left(\frac{TP}{TP+FN} + \frac{TN}{TN+FP}\right)$, where $TN$ is the number of true negatives. Balanced Accuracy assigns equal weight to the recall of each class, preventing dominance by the majority class.

Table 6: Summary statistics of the CLIMB benchmark datasets, grouped by imbalance level.

| Imbalance level | Statistic | samples | features | IR |
|---|---|---|---|---|
| Low (28 datasets) | min | 161 | 3 | 2.01 |
| | max | 175,028 | 107 | 3.95 |
| | mean | 22,562.21 | 24.5 | 2.67 |
| Medium (24 datasets) | min | 327 | 4 | 5.17 |
| | max | 101,766 | 117 | 9.92 |
| | mean | 11,885.71 | 37.92 | 7.3 |
| High (15 datasets) | min | 294 | 4 | 10 |
| | max | 149,639 | 85 | 42.01 |
| | mean | 35,001.8 | 19.91 | 27.4 |
| Extreme (6 datasets) | min | 3,772 | 9 | 64.03 |
| | max | 284,807 | 36 | 577.88 |
| | mean | 56,229.33 | 23.83 | 201.82 |

Table 7: Summary statistics of datasets from the *imbalanced-learn* package.

| Dataset | samples | features | IR |
|---|---|---|---|
| optical_digits | 5,620 | 64 | 9.14 |
| satimage | 6,435 | 36 | 9.28 |
| pen_digits | 10,992 | 16 | 9.42 |
| sick_euthyroid | 3,163 | 42 | 9.8 |
| thyroid_sick | 3,772 | 52 | 15.33 |
| coil_2000 | 9,822 | 85 | 15.76 |
| wine_quality | 4,898 | 11 | 25.77 |
| letter_img | 20,000 | 16 | 26.25 |
| abalone_19 | 4,177 | 10 | 129.53 |
| isolet | 7,797 | 617 | 11.99 |

- F1-Score: $2 \cdot \frac{Precision \cdot Recall}{Precision + Recall}$, which is the harmonic mean of precision and recall, providing a single metric that balances both.

- G-mean: $\sqrt{Precision \cdot Recall}$, which is the geometric mean of precision and recall, providing a single metric that balances both, and is particularly useful in imbalanced datasets as it penalizes low values in either precision or recall.

- MCC: $\frac{TP \cdot TN - FP \cdot FN}{\sqrt{(TP+FP)(TP+FN)(TN+FP)(TN+FN)}}$, which considers all four entries of the confusion matrix and therefore provides a more balanced evaluation for imbalanced classification problems.

## C results

### C.1 Results of SMOTE-Based Methods

Table 8: Performance comparison (mean ± standard error) among SMOTE-based minority data augmentation on AUPRC.

| Classifier | Oversampling Method | Imbalance Level | | | | |
| --- | --- | --- | --- | --- | --- | --- |
| | | Low | Medium | High | Extreme | Average |
| DecisionTree | None | 50.49±0.20 | 50.90±0.23 | 34.02±0.30 | 42.90±1.05 | 46.62±0.15 |
| | SMOTE | 50.86±0.20 | 50.09±0.20 | 33.35±0.26 | 34.08±0.76 | 45.63±0.13 |
| | BorderlineSMOTE | 50.07±0.21 | 50.12±0.22 | 33.24±0.22 | 37.10±0.80 | 45.56±0.13 |
| | SVMSMOTE | 50.49±0.21 | 50.73±0.24 | 33.78±0.22 | 38.55±0.63 | 46.15±0.13 |
| | SMOTE-TomekLinks | 50.94±0.22 | 50.79±0.23 | 32.73±0.28 | 34.91±0.59 | 45.83±0.14 |
| | TreeSMOTE | 51.26±0.20 | 51.71±0.23 | 34.47±0.23 | 42.89±0.97 | 47.27±0.14 |
| LinearSVC | None | 68.56±0.25 | 66.33±0.18 | 42.10±0.29 | 54.06±0.59 | 61.20±0.14 |
| | SMOTE | 68.30±0.26 | 65.18±0.21 | 40.51±0.33 | 52.53±0.61 | 60.27±0.15 |
| | BorderlineSMOTE | 67.94±0.25 | 64.33±0.21 | 37.93±0.32 | 52.44±0.68 | 59.31±0.15 |
| | SVMSMOTE | 68.06±0.24 | 65.28±0.20 | 39.62±0.33 | 53.20±0.69 | 60.08±0.14 |
| | SMOTE-TomekLinks | 68.35±0.26 | 65.65±0.19 | 40.95±0.33 | 52.13±0.65 | 60.50±0.14 |
| | TreeSMOTE | 68.60±0.25 | 65.59±0.20 | 41.13±0.32 | 53.63±0.62 | 60.73±0.14 |
| MLP | None | 71.49±0.22 | 70.68±0.18 | 52.46±0.27 | 63.02±0.75 | 66.62±0.13 |
| | SMOTE | 71.03±0.24 | 69.62±0.19 | 51.28±0.31 | 61.19±0.66 | 65.70±0.14 |
| | BorderlineSMOTE | 69.87±0.22 | 69.41±0.18 | 51.27±0.31 | 61.81±0.70 | 65.23±0.13 |
| | SVMSMOTE | 70.42±0.21 | 69.94±0.19 | 52.21±0.30 | 62.09±0.76 | 65.84±0.13 |
| | SMOTE-TomekLinks | 70.91±0.25 | 69.64±0.19 | 51.15±0.33 | 61.58±0.68 | 65.67±0.14 |
| | TreeSMOTE | 71.14±0.21 | 70.18±0.19 | 52.03±0.26 | 63.75±0.84 | 66.29±0.13 |
| TabM | None | 56.39±0.28 | 62.77±0.19 | 46.10±0.28 | 45.64±0.60 | 55.62±0.14 |
| | SMOTE | 56.83±0.25 | 62.09±0.19 | 46.13±0.33 | 48.55±0.88 | 55.81±0.15 |
| | BorderlineSMOTE | 56.42±0.21 | 62.25±0.21 | 47.11±0.39 | 49.62±0.83 | 55.99±0.15 |
| | SVMSMOTE | 56.81±0.20 | 62.76±0.22 | 47.24±0.31 | 50.52±0.85 | 56.41±0.14 |
| | SMOTE-TomekLinks | 56.95±0.23 | 62.32±0.20 | 45.77±0.33 | 48.65±1.00 | 55.87±0.15 |
| | TreeSMOTE | 57.36±0.22 | 63.30±0.19 | 47.94±0.29 | 52.61±1.00 | 57.11±0.14 |
| LightGBM | None | 72.35±0.19 | 73.31±0.18 | 57.67±0.35 | 54.79±1.26 | 68.35±0.14 |
| | SMOTE | 72.17±0.20 | 73.12±0.17 | 57.93±0.30 | 65.94±0.91 | 69.20±0.13 |
| | BorderlineSMOTE | 72.00±0.18 | 72.81±0.17 | 58.11±0.25 | 60.38±1.20 | 68.60±0.13 |
| | SVMSMOTE | 72.38±0.22 | 73.16±0.17 | 58.42±0.28 | 62.85±1.18 | 69.13±0.14 |
| | SMOTE-TomekLinks | 71.85±0.17 | 73.27±0.18 | 58.17±0.23 | 61.61±1.10 | 68.81±0.12 |
| | TreeSMOTE | 72.37±0.20 | 73.39±0.17 | 58.74±0.27 | 66.65±1.40 | 69.58±0.14 |

Table 9: Performance comparison (mean ± standard error) among SMOTE-based minority data augmentation on BAC.

| Classifier | Oversampling Method | Imbalance Level | | | | |
|---|---|---|---|---|---|---|
| | | Low | Medium | High | Extreme | Average |
| DecisionTree | None | 71.71±0.17 | 74.45±0.18 | 60.86±0.23 | 75.19±0.61 | 70.67±0.11 |
| | SMOTE | 72.71±0.18 | 75.35±0.16 | 63.02±0.27 | 75.61±0.83 | 71.83±0.12 |
| | BorderlineSMOTE | 72.23±0.19 | 75.24±0.19 | 61.95±0.21 | 73.81±0.66 | 71.24±0.12 |
| | SVMSMOTE | 72.39±0.19 | 75.74±0.19 | 62.74±0.21 | 74.65±0.51 | 71.70±0.11 |
| | SMOTE-TomekLinks | 72.97±0.18 | 75.44±0.17 | 62.07±0.25 | 75.54±0.52 | 71.75±0.11 |
| | TreeSMOTE | 72.63±0.18 | 75.57±0.16 | 61.85±0.15 | 75.63±0.83 | 71.63±0.11 |
| LinearSVC | None | 71.67±0.14 | 70.16±0.11 | 51.43±0.15 | 62.98±0.31 | 66.30±0.08 |
| | SMOTE | 76.27±0.19 | 78.87±0.15 | 66.09±0.23 | 82.32±0.69 | 75.53±0.11 |
| | BorderlineSMOTE | 75.96±0.19 | 78.24±0.15 | 65.09±0.20 | 80.71±0.64 | 74.87±0.11 |
| | SVMSMOTE | 76.01±0.19 | 78.34±0.15 | 64.41±0.22 | 81.01±0.67 | 74.80±0.11 |
| | SMOTE-TomekLinks | 76.39±0.20 | 76.41±0.15 | 62.56±0.19 | 74.31±0.70 | 73.38±0.11 |
| | TreeSMOTE | 76.24±0.18 | 78.20±0.14 | 62.83±0.19 | 79.34±0.61 | 74.39±0.10 |
| MLP | None | 75.07±0.15 | 75.76±0.13 | 61.88±0.22 | 69.95±0.62 | 72.17±0.09 |
| | SMOTE | 77.82±0.18 | 78.96±0.14 | 67.80±0.23 | 78.81±0.66 | 76.22±0.11 |
| | BorderlineSMOTE | 77.61±0.16 | 78.99±0.15 | 67.25±0.20 | 78.06±0.70 | 75.97±0.10 |
| | SVMSMOTE | 77.77±0.17 | 79.16±0.15 | 67.37±0.21 | 77.59±0.77 | 76.08±0.11 |
| | SMOTE-TomekLinks | 77.85±0.18 | 77.87±0.15 | 65.61±0.23 | 75.27±0.66 | 75.13±0.11 |
| | TreeSMOTE | 77.61±0.15 | 78.66±0.14 | 66.68±0.23 | 77.77±0.65 | 75.72±0.10 |
| TabM | None | 74.31±0.22 | 76.33±0.14 | 64.73±0.20 | 67.20±0.51 | 72.53±0.11 |
| | SMOTE | 77.46±0.22 | 78.93±0.16 | 70.47±0.27 | 76.39±0.90 | 76.50±0.13 |
| | BorderlineSMOTE | 77.61±0.18 | 78.79±0.18 | 70.45±0.26 | 75.19±0.91 | 76.41±0.12 |
| | SVMSMOTE | 77.66±0.16 | 79.41±0.16 | 70.20±0.22 | 75.90±0.88 | 76.65±0.11 |
| | SMOTE-TomekLinks | 77.54±0.19 | 78.46±0.16 | 69.01±0.21 | 73.05±0.79 | 75.82±0.12 |
| | TreeSMOTE | 77.17±0.18 | 79.25±0.15 | 70.20±0.20 | 75.81±0.84 | 76.39±0.11 |
| LightGBM | None | 75.32±0.14 | 76.87±0.16 | 65.64±0.27 | 69.51±0.98 | 73.47±0.11 |
| | SMOTE | 76.97±0.17 | 78.77±0.14 | 70.58±0.23 | 78.40±0.64 | 76.45±0.10 |
| | BorderlineSMOTE | 77.24±0.18 | 78.62±0.13 | 70.34±0.22 | 78.17±0.84 | 76.44±0.11 |
| | SVMSMOTE | 77.11±0.17 | 79.08±0.13 | 69.91±0.23 | 77.31±0.60 | 76.38±0.10 |
| | SMOTE-TomekLinks | 77.24±0.17 | 78.48±0.18 | 68.78±0.22 | 75.03±0.65 | 75.82±0.11 |
| | TreeSMOTE | 76.58±0.17 | 78.49±0.14 | 69.15±0.22 | 77.78±0.66 | 75.87±0.10 |

Table 10: Performance comparison (mean ± standard error) among SMOTE-based minority data augmentation on F1-Score.

| Classifier | Oversampling Method | Imbalance Level | | | | |
| --- | --- | --- | --- | --- | --- | --- |
| | | Low | Medium | High | Extreme | Average |
| Classification Tree | None | 71.65±0.16 | 74.23±0.17 | 60.76±0.20 | 75.30±0.55 | 70.56±0.10 |
| | SMOTE | 72.17±0.17 | 74.09±0.14 | 60.57±0.20 | 70.19±0.63 | 70.26±0.10 |
| | BorderlineSMOTE | 71.60±0.18 | 74.16±0.17 | 60.61±0.17 | 71.80±0.54 | 70.20±0.10 |
| | SVMSMOTE | 71.85±0.18 | 74.42±0.18 | 61.08±0.17 | 72.48±0.39 | 70.53±0.10 |
| | SMOTE-TomekLinks | 72.40±0.18 | 74.31±0.16 | 60.28±0.21 | 71.16±0.43 | 70.44±0.10 |
| | TreeSMOTE | 72.36±0.17 | 74.81±0.16 | 61.40±0.14 | 74.77±0.56 | 71.11±0.10 |
| LinearSVC | None | 71.62±0.14 | 71.15±0.13 | 52.52±0.16 | 66.38±0.24 | 67.11±0.08 |
| | SMOTE | 73.68±0.18 | 73.00±0.13 | 52.62±0.14 | 57.11±0.25 | 67.77±0.09 |
| | BorderlineSMOTE | 72.41±0.18 | 71.46±0.15 | 52.50±0.16 | 60.04±0.36 | 66.99±0.09 |
| | SVMSMOTE | 73.20±0.19 | 73.14±0.15 | 53.66±0.21 | 62.67±0.38 | 68.30±0.10 |
| | SMOTE-TomekLinks | 73.77±0.19 | 71.58±0.14 | 50.84±0.13 | 56.80±0.24 | 66.94±0.09 |
| | TreeSMOTE | 74.78±0.18 | 74.80±0.13 | 55.71±0.17 | 66.95±0.35 | 70.23±0.09 |
| MLP | None | 75.03±0.15 | 76.59±0.13 | 63.00±0.22 | 72.30±0.59 | 72.85±0.10 |
| | SMOTE | 75.99±0.18 | 77.01±0.13 | 62.90±0.20 | 73.46±0.37 | 73.43±0.10 |
| | BorderlineSMOTE | 75.20±0.16 | 76.91±0.14 | 63.54±0.19 | 74.44±0.45 | 73.30±0.09 |
| | SVMSMOTE | 75.87±0.18 | 77.56±0.14 | 64.36±0.21 | 74.20±0.49 | 73.93±0.10 |
| | SMOTE-TomekLinks | 76.05±0.18 | 76.46±0.13 | 62.16±0.21 | 73.10±0.45 | 73.09±0.10 |
| | TreeSMOTE | 76.80±0.16 | 77.77±0.13 | 64.84±0.21 | 75.59±0.43 | 74.56±0.09 |
| TabM | None | 73.97±0.25 | 76.88±0.14 | 65.29±0.19 | 68.77±0.41 | 72.82±0.11 |
| | SMOTE | 76.36±0.21 | 77.95±0.14 | 68.29±0.21 | 74.23±0.66 | 75.15±0.11 |
| | BorderlineSMOTE | 76.16±0.16 | 77.94±0.14 | 69.17±0.24 | 74.89±0.65 | 75.29±0.10 |
| | SVMSMOTE | 76.43±0.15 | 78.65±0.16 | 69.16±0.20 | 75.17±0.66 | 75.65±0.10 |
| | SMOTE-TomekLinks | 76.41±0.19 | 77.79±0.14 | 67.25±0.18 | 73.14±0.56 | 74.82±0.10 |
| | TreeSMOTE | 76.77±0.17 | 78.86±0.14 | 69.87±0.18 | 76.18±0.68 | 76.08±0.10 |
| LightGBM | None | 75.90±0.15 | 78.04±0.15 | 67.06±0.30 | 71.83±0.84 | 74.55±0.11 |
| | SMOTE | 76.93±0.17 | 78.86±0.12 | 68.38±0.21 | 76.28±0.46 | 75.86±0.10 |
| | BorderlineSMOTE | 76.86±0.18 | 78.71±0.12 | 68.81±0.20 | 76.96±0.71 | 75.92±0.10 |
| | SVMSMOTE | 76.79±0.17 | 79.06±0.12 | 68.99±0.23 | 77.02±0.54 | 76.05±0.10 |
| | SMOTE-TomekLinks | 77.13±0.17 | 78.88±0.17 | 67.86±0.21 | 73.19±0.50 | 75.59±0.10 |
| | TreeSMOTE | 76.71±0.17 | 79.06±0.13 | 69.46±0.21 | 79.11±0.51 | 76.28±0.10 |

Table 11: Performance comparison (mean ± standard error) among SMOTE-based minority data augmentation on G-mean.

| Classifier | Oversampling Method | Imbalance Level | | | | |
| --- | --- | --- | --- | --- | --- | --- |
| | | Low | Medium | High | Extreme | Average |
| DecisionTree | None | 71.71±0.16 | 74.31±0.17 | 60.80±0.20 | 75.53±0.56 | 70.64±0.10 |
| | SMOTE | 72.27±0.17 | 74.22±0.14 | 60.92±0.20 | 70.90±0.64 | 70.46±0.10 |
| | BorderlineSMOTE | 71.68±0.18 | 74.26±0.17 | 60.79±0.17 | 72.12±0.56 | 70.33±0.10 |
| | SVMSMOTE | 71.93±0.18 | 74.52±0.18 | 61.26±0.17 | 72.85±0.40 | 70.66±0.10 |
| | SMOTE-TomekLinks | 72.50±0.18 | 74.46±0.17 | 60.54±0.21 | 71.82±0.42 | 70.63±0.10 |
| | TreeSMOTE | 72.42±0.17 | 74.89±0.16 | 61.46±0.14 | 75.09±0.58 | 71.20±0.10 |
| LinearSVC | None | 72.57±0.15 | 72.38±0.14 | 53.68±0.16 | 67.13±0.21 | 68.18±0.08 |
| | SMOTE | 74.30±0.18 | 74.20±0.13 | 55.57±0.14 | 61.46±0.25 | 69.36±0.09 |
| | BorderlineSMOTE | 73.33±0.18 | 72.82±0.14 | 55.25±0.15 | 63.61±0.33 | 68.65±0.09 |
| | SVMSMOTE | 73.83±0.19 | 73.96±0.14 | 55.84±0.19 | 65.69±0.35 | 69.51±0.10 |
| | SMOTE-TomekLinks | 74.40±0.19 | 73.06±0.14 | 53.59±0.13 | 60.36±0.27 | 68.53±0.09 |
| | TreeSMOTE | 75.10±0.18 | 75.32±0.13 | 56.98±0.17 | 68.79±0.36 | 70.93±0.09 |
| MLP | None | 75.62±0.16 | 77.11±0.13 | 63.83±0.24 | 72.99±0.60 | 73.47±0.10 |
| | SMOTE | 76.41±0.18 | 77.33±0.13 | 63.82±0.20 | 74.21±0.38 | 73.94±0.09 |
| | BorderlineSMOTE | 75.74±0.16 | 77.24±0.14 | 64.18±0.18 | 74.91±0.46 | 73.79±0.09 |
| | SVMSMOTE | 76.19±0.17 | 77.74±0.14 | 64.81±0.20 | 74.60±0.50 | 74.23±0.10 |
| | SMOTE-TomekLinks | 76.46±0.18 | 76.89±0.13 | 63.14±0.21 | 73.69±0.46 | 73.64±0.10 |
| | TreeSMOTE | 76.99±0.15 | 77.91±0.12 | 65.12±0.21 | 75.85±0.45 | 74.76±0.09 |
| TabM | None | 74.56±0.24 | 77.48±0.14 | 66.08±0.18 | 69.36±0.36 | 73.45±0.11 |
| | SMOTE | 76.62±0.21 | 78.21±0.14 | 68.61±0.22 | 74.58±0.66 | 75.42±0.11 |
| | BorderlineSMOTE | 76.47±0.16 | 78.22±0.15 | 69.36±0.24 | 75.17±0.66 | 75.56±0.11 |
| | SVMSMOTE | 76.61±0.15 | 78.83±0.16 | 69.30±0.19 | 75.44±0.67 | 75.83±0.10 |
| | SMOTE-TomekLinks | 76.64±0.19 | 78.17±0.14 | 67.82±0.18 | 73.56±0.54 | 75.18±0.10 |
| | TreeSMOTE | 76.88±0.17 | 79.06±0.14 | 70.04±0.18 | 76.44±0.66 | 76.24±0.10 |
| LightGBM | None | 76.28±0.15 | 78.47±0.15 | 68.00±0.30 | 72.93±0.83 | 75.12±0.11 |
| | SMOTE | 77.03±0.17 | 79.06±0.12 | 68.84±0.21 | 76.96±0.45 | 76.11±0.09 |
| | BorderlineSMOTE | 76.97±0.18 | 78.93±0.12 | 69.18±0.20 | 77.47±0.70 | 76.15±0.10 |
| | SVMSMOTE | 76.90±0.17 | 79.34±0.12 | 69.34±0.23 | 77.52±0.53 | 76.29±0.10 |
| | SMOTE-TomekLinks | 77.24±0.17 | 79.21±0.16 | 68.46±0.21 | 73.95±0.51 | 75.91±0.10 |
| | TreeSMOTE | 76.82±0.17 | 79.30±0.14 | 69.85±0.22 | 79.93±0.50 | 76.55±0.10 |

Table 12: Performance comparison (mean ± standard error) among SMOTE-based minority data augmentation on MCC.

| Classifier | Oversampling Method | Imbalance Level | | | | |
|---|---|---|---|---|---|---|
| | | Low | Medium | High | Extreme | Average |
| DecisionTree | None | 43.52±0.33 | 56.04±0.28 | 41.92±0.36 | 56.17±0.92 | 48.35±0.19 |
| | SMOTE | 44.69±0.34 | 55.74±0.25 | 41.52±0.35 | 49.55±0.80 | 48.07±0.18 |
| | BorderlineSMOTE | 43.51±0.36 | 56.07±0.29 | 41.37±0.26 | 51.71±0.82 | 47.87±0.19 |
| | SVMSMOTE | 43.98±0.37 | 56.39±0.31 | 42.30±0.29 | 52.97±0.79 | 48.45±0.19 |
| | SMOTE-TomekLinks | 45.16±0.35 | 56.88±0.27 | 41.45±0.33 | 50.28±0.56 | 48.67±0.18 |
| | TreeSMOTE | 44.96±0.35 | 57.19±0.26 | 42.58±0.29 | 57.47±0.87 | 49.52±0.18 |
| LinearSVC | None | 47.12±0.31 | 55.31±0.23 | 33.32±0.26 | 45.71±0.46 | 46.86±0.16 |
| | SMOTE | 49.74±0.36 | 57.37±0.22 | 35.94±0.24 | 34.53±0.44 | 48.16±0.16 |
| | BorderlineSMOTE | 48.38±0.35 | 55.16±0.21 | 34.73±0.28 | 37.39±0.59 | 46.90±0.17 |
| | SVMSMOTE | 48.80±0.37 | 56.21±0.23 | 35.61±0.35 | 40.56±0.59 | 47.85±0.18 |
| | SMOTE-TomekLinks | 49.96±0.38 | 57.76±0.22 | 36.03±0.23 | 37.06±0.56 | 48.60±0.17 |
| | TreeSMOTE | 50.76±0.35 | 58.37±0.21 | 37.15±0.30 | 45.37±0.54 | 50.02±0.16 |
| MLP | None | 52.58±0.31 | 61.92±0.21 | 47.29±0.38 | 54.21±0.80 | 54.70±0.17 |
| | SMOTE | 53.60±0.35 | 61.65±0.21 | 46.57±0.30 | 55.72±0.64 | 54.98±0.17 |
| | BorderlineSMOTE | 52.48±0.32 | 61.48±0.22 | 47.53±0.29 | 56.83±0.83 | 54.78±0.16 |
| | SVMSMOTE | 52.95±0.33 | 62.18±0.23 | 48.28±0.32 | 56.96±0.81 | 55.35±0.17 |
| | SMOTE-TomekLinks | 53.66±0.36 | 61.66±0.21 | 46.82±0.30 | 56.11±0.63 | 55.09±0.17 |
| | TreeSMOTE | 54.32±0.30 | 62.59±0.20 | 48.43±0.34 | 58.47±0.70 | 56.17±0.16 |
| TabM | None | 51.48±0.43 | 62.40±0.23 | 46.68±0.33 | 49.10±0.49 | 53.99±0.19 |
| | SMOTE | 53.69±0.41 | 62.83±0.24 | 49.99±0.37 | 56.70±0.87 | 56.27±0.20 |
| | BorderlineSMOTE | 53.47±0.33 | 63.05±0.25 | 51.46±0.39 | 57.90±0.78 | 56.64±0.18 |
| | SVMSMOTE | 53.54±0.30 | 63.84±0.27 | 51.41±0.31 | 58.55±0.87 | 56.97±0.17 |
| | SMOTE-TomekLinks | 53.70±0.37 | 63.23±0.24 | 50.57±0.35 | 57.11±0.89 | 56.55±0.19 |
| | TreeSMOTE | 54.03±0.34 | 64.26±0.24 | 52.12±0.27 | 60.50±0.88 | 57.61±0.17 |
| LightGBM | None | 53.31±0.31 | 63.45±0.25 | 50.81±0.44 | 52.47±1.06 | 56.14±0.19 |
| | SMOTE | 54.21±0.34 | 64.48±0.22 | 51.25±0.31 | 60.60±0.56 | 57.59±0.17 |
| | BorderlineSMOTE | 54.13±0.35 | 64.13±0.21 | 51.80±0.32 | 61.88±0.92 | 57.66±0.17 |
| | SVMSMOTE | 53.96±0.34 | 65.02±0.23 | 51.82±0.33 | 62.95±0.69 | 57.98±0.17 |
| | SMOTE-TomekLinks | 54.63±0.34 | 65.00±0.26 | 51.16±0.33 | 58.00±0.88 | 57.69±0.18 |
| | TreeSMOTE | 53.81±0.34 | 64.88±0.25 | 52.83±0.32 | 65.79±0.79 | 58.31±0.18 |

## C.2 Results of Generative Model-Based Methods

Table 13: AUPRC comparison (mean ± standard error) of TreeSMOTE and generative model-based oversampling methods on 10 datasets.

| Dataset | TVAE | CTGAN | CTGANENN | DGOT | TabDDPM | TreeSMOTE |
|---|---|---|---|---|---|---|
| optical_digits | 83.66±0.20 | 77.49±0.25 | 87.64±0.36 | 89.50±0.23 | 89.29±0.13 | 91.05±0.24 |
| satimage | 38.52±0.43 | 38.77±0.41 | 35.59±0.32 | 40.95±0.37 | 40.30±0.39 | 43.12±0.48 |
| pen_digits | 90.31±0.21 | 89.18±0.11 | 88.72±0.38 | 93.45±0.24 | 90.84±0.11 | 93.92±0.17 |
| sick_euthyroid | 64.98±0.92 | 50.38±0.70 | 51.10±2.44 | 77.12±0.61 | 74.21±0.49 | 76.39±0.49 |
| thyroid_sick | 58.55±1.18 | 48.58±1.06 | 48.13±1.62 | 75.07±1.15 | 73.24±0.61 | 76.76±0.50 |
| coil_2000 | 10.82±0.11 | 9.62±0.19 | 9.28±0.13 | 11.07±0.10 | 10.83±0.07 | 11.03±0.09 |
| wine_quality | 23.22±0.76 | 17.09±0.54 | 15.44±0.54 | 21.92±0.78 | 21.73±0.70 | 25.20±1.07 |
| letter_img | 81.82±0.80 | 81.94±0.40 | 83.67±0.24 | 91.57±0.14 | 86.39±0.20 | 90.55±0.19 |
| abalone_19 | 1.92±0.18 | 2.86±0.34 | 1.99±0.14 | 1.72±0.04 | 2.70±0.31 | 3.66±0.50 |
| isolet | 79.00±0.23 | 80.54±0.20 | 77.91±0.20 | 79.50±0.17 | 80.71±0.11 | 82.29±0.18 |

Table 14: BAC comparison (mean ± standard error) of TreeSMOTE and generative model-based oversampling methods on 10 datasets.

| Dataset | TVAE | CTGAN | CTGANENN | DGOT | TabDDPM | TreeSMOTE |
|---|---|---|---|---|---|---|
| optical_digits | 92.84±0.20 | 91.91±0.11 | 93.88±0.19 | 93.37±0.16 | 94.10±0.10 | 95.26±0.17 |
| satimage | 72.16±0.29 | 67.44±0.15 | 62.29±0.35 | 66.50±0.53 | 72.18±0.33 | 75.92±0.35 |
| pen_digits | 97.11±0.04 | 96.48±0.05 | 94.15±0.10 | 95.62±0.10 | 96.12±0.06 | 97.42±0.09 |
| sick_euthyroid | 90.07±0.27 | 81.29±0.43 | 70.85±0.98 | 85.99±0.44 | 88.94±0.22 | 92.01±0.22 |
| thyroid_sick | 87.01±0.35 | 79.23±0.50 | 68.27±0.77 | 80.94±0.59 | 87.20±0.20 | 90.44±0.15 |
| coil_2000 | 53.37±0.16 | 52.49±0.27 | 51.71±0.11 | 51.77±0.07 | 53.95±0.09 | 55.35±0.13 |
| wine_quality | 65.79±0.24 | 68.06±0.55 | 58.97±0.38 | 56.19±0.29 | 62.47±0.34 | 67.09±0.44 |
| letter_img | 94.53±0.24 | 96.10±0.09 | 95.39±0.10 | 93.80±0.11 | 94.26±0.10 | 97.06±0.10 |
| abalone_19 | 55.07±1.03 | 54.23±0.53 | 52.19±0.33 | 50.15±0.10 | 50.81±0.14 | 50.82±0.29 |
| isolet | 90.77±0.11 | 90.50±0.07 | 84.49±0.16 | 89.06±0.11 | 90.63±0.12 | 91.50±0.11 |

Table 15: F1-Score comparison (mean ± standard error) among minority data augmentation by a generative model and TreeSMOTE.

| Dataset | TVAE | CTGAN | CTGANENN | DGOT | TabDDPM | TreeSMOTE |
|---|---|---|---|---|---|---|
| optical_digits | 91.03±0.11 | 88.48±0.12 | 93.57±0.16 | 94.28±0.13 | 94.16±0.09 | 94.53±0.13 |
| satimage | 68.74±0.22 | 67.53±0.11 | 64.49±0.34 | 66.45±0.29 | 68.40±0.26 | 70.40±0.31 |
| pen_digits | 92.81±0.10 | 93.10±0.04 | 93.19±0.10 | 96.14±0.15 | 94.69±0.04 | 96.01±0.06 |
| sick_euthyroid | 81.79±0.66 | 76.52±0.41 | 74.41±1.06 | 87.61±0.34 | 87.00±0.23 | 87.88±0.31 |
| thyroid_sick | 79.14±0.39 | 75.14±0.50 | 72.34±0.82 | 83.05±0.61 | 84.90±0.29 | 86.82±0.12 |
| coil_2000 | 53.40±0.19 | 51.44±0.25 | 50.28±0.08 | 51.18±0.06 | 52.94±0.07 | 54.43±0.10 |
| wine_quality | 65.02±0.32 | 58.65±0.48 | 59.61±0.32 | 56.82±0.33 | 62.01±0.35 | 65.97±0.49 |
| letter_img | 88.19±0.28 | 88.36±0.30 | 90.27±0.12 | 95.17±0.07 | 91.90±0.09 | 93.56±0.07 |
| abalone_19 | 48.34±0.11 | 50.07±0.27 | 50.05±0.07 | 49.94±0.07 | 50.36±0.07 | 50.98±0.42 |
| isolet | 89.42±0.10 | 90.29±0.08 | 87.82±0.07 | 89.68±0.09 | 90.66±0.06 | 91.22±0.08 |

Table 16: G-mean comparison (mean ± standard error) of TreeSMOTE and generative model-based oversampling methods on 10 datasets.

| Dataset | TVAE | CTGAN | CTGANENN | DGOT | TabDDPM | TreeSMOTE |
|---|---|---|---|---|---|---|
| optical_digits | 91.16±0.10 | 88.93±0.11 | 93.61±0.15 | 94.32±0.13 | 94.20±0.09 | 94.62±0.13 |
| satimage | 69.80±0.20 | 68.15±0.15 | 65.98±0.26 | 67.63±0.36 | 70.01±0.26 | 72.06±0.25 |
| pen_digits | 93.34±0.09 | 93.50±0.03 | 93.26±0.09 | 96.17±0.15 | 94.87±0.04 | 96.16±0.06 |
| sick_euthyroid | 82.87±0.58 | 77.08±0.41 | 75.22±1.07 | 88.04±0.31 | 87.38±0.22 | 88.31±0.29 |
| thyroid_sick | 80.07±0.37 | 75.88±0.46 | 73.82±0.81 | 83.81±0.65 | 85.46±0.28 | 87.36±0.12 |
| coil_2000 | 53.86±0.16 | 52.23±0.31 | 50.45±0.09 | 51.75±0.06 | 53.55±0.10 | 55.09±0.09 |
| wine_quality | 65.40±0.34 | 60.57±0.46 | 60.26±0.33 | 58.25±0.36 | 62.55±0.34 | 66.43±0.48 |
| letter_img | 88.97±0.24 | 89.30±0.25 | 91.04±0.11 | 95.29±0.07 | 92.31±0.08 | 94.08±0.07 |
| abalone_19 | 49.67±0.24 | 50.90±0.29 | 50.45±0.09 | 49.95±0.07 | 50.46±0.07 | 51.35±0.56 |
| isolet | 89.53±0.10 | 90.34±0.08 | 88.12±0.07 | 89.71±0.09 | 90.69±0.06 | 91.24±0.08 |

Table 17: MCC comparison (mean ± standard error) of TreeSMOTE and generative model-based oversampling methods on 10 datasets.

| Dataset | TVAE | CTGAN | CTGANENN | DGOT | TabDDPM | TreeSMOTE |
|---|---|---|---|---|---|---|
| optical_digits | 82.39±0.21 | 78.21±0.21 | 87.25±0.30 | 88.65±0.25 | 88.41±0.18 | 89.29±0.27 |
| satimage | 41.34±0.41 | 37.24±0.33 | 33.58±0.46 | 37.25±0.73 | 42.43±0.52 | 46.96±0.42 |
| pen_digits | 87.12±0.17 | 87.30±0.05 | 86.56±0.17 | 92.35±0.30 | 89.86±0.08 | 92.43±0.13 |
| sick_euthyroid | 66.58±1.07 | 54.65±0.80 | 50.79±2.13 | 76.24±0.60 | 75.00±0.43 | 76.91±0.56 |
| thyroid_sick | 60.63±0.74 | 52.35±0.87 | 48.08±1.61 | 68.10±1.27 | 71.93±0.30 | 75.00±0.25 |
| coil_2000 | 8.15±0.30 | 5.50±0.62 | 2.84±0.18 | 4.89±0.14 | 7.54±0.11 | 10.70±0.17 |
| wine_quality | 30.92±0.69 | 23.17±0.87 | 20.74±0.67 | 17.51±0.72 | 25.45±0.67 | 33.04±0.95 |
| letter_img | 78.26±0.46 | 78.96±0.47 | 82.41±0.22 | 90.61±0.15 | 84.81±0.17 | 88.32±0.14 |
| abalone_19 | 2.64±0.60 | 3.20±0.53 | 1.35±0.17 | 0.20±0.15 | 1.69±0.50 | 2.80±1.13 |
| isolet | 79.11±0.20 | 80.69±0.16 | 76.34±0.14 | 79.43±0.17 | 81.39±0.11 | 82.48±0.17 |

### C.3 Results of Ensemble-Based Methods

Table 18: Performance comparison (mean ± standard error) between ensemble methods and their combinations with TreeSMOTE across different imbalance levels (low, medium, high, and extreme) on AUPRC.

| Method | Low IRs | Medium IRs | High IRs | Extreme IRs | Average |
|---|---|---|---|---|---|
| SelfPacedEnsemble | 71.55±0.21 | 71.65±0.16 | 54.66±0.21 | 64.86±0.88 | 67.39±0.12 |
| SelfPacedEnsemble w/ TreeSmote | 71.67±0.22 | 72.16±0.15 | 54.68±0.17 | 65.07±0.83 | 67.63±0.12 |
| BalancedRandomForest | 71.40±0.25 | 70.46±0.17 | 51.89±0.30 | 59.09±0.99 | 65.84±0.14 |
| BalancedRandomForest w/ TreeSmote | 71.04±0.24 | 70.23±0.20 | 51.97±0.28 | 58.95±0.99 | 65.64±0.15 |
| RUSBoost | 71.09±0.25 | 69.31±0.20 | 49.85±0.27 | 56.05±1.03 | 64.64±0.15 |
| RUSBoost w/ TreeSmote | 70.79±0.22 | 69.98±0.21 | 50.54±0.29 | 56.54±0.98 | 64.95±0.14 |
| UnderBagging | 70.94±0.23 | 70.50±0.18 | 52.96±0.27 | 59.35±1.14 | 65.94±0.14 |
| UnderBagging w/ TreeSmote | 71.03±0.25 | 70.33±0.20 | 52.70±0.28 | 58.40±0.95 | 65.78±0.14 |
| BalanceCascade | 70.69±0.22 | 70.55±0.18 | 51.03±0.27 | 63.00±0.86 | 65.77±0.13 |
| BalanceCascade w/ TreeSmote | 70.88±0.25 | 70.49±0.19 | 50.99±0.25 | 63.00±0.85 | 65.81±0.14 |

Table 19: Performance comparison (mean ± standard error) between ensemble methods and their combinations with TreeSMOTE across different imbalance levels (low, medium, high, and extreme) on BAC.

| Method | Low IRs | Medium IRs | High IRs | Extreme IRs | Average |
|---|---|---|---|---|---|
| SelfPacedEnsemble | 77.21±0.18 | 80.83±0.16 | 69.66±0.20 | 85.03±0.52 | 77.51±0.10 |
| SelfPacedEnsemble w/ TreeSmote | 77.11±0.19 | 80.97±0.15 | 69.92±0.19 | 85.04±0.57 | 77.57±0.11 |
| BalancedRandomForest | 78.38±0.18 | 82.22±0.12 | 71.33±0.18 | 86.17±0.48 | 78.85±0.09 |
| BalancedRandomForest w/ TreeSmote | 78.19±0.19 | 82.34±0.13 | 71.51±0.16 | 86.35±0.54 | 78.88±0.10 |
| RUSBoost | 77.00±0.20 | 80.29±0.13 | 68.61±0.20 | 84.16±0.61 | 76.94±0.10 |
| RUSBoost w/ TreeSmote | 77.35±0.19 | 80.81±0.14 | 69.53±0.23 | 83.80±0.67 | 77.41±0.11 |
| UnderBagging | 77.64±0.19 | 81.75±0.15 | 71.19±0.15 | 85.96±0.46 | 78.38±0.10 |
| UnderBagging w/ TreeSmote | 77.60±0.18 | 81.75±0.13 | 71.45±0.21 | 86.40±0.47 | 78.46±0.10 |
| BalanceCascade | 76.56±0.19 | 80.27±0.14 | 66.48±0.24 | 83.23±0.64 | 76.24±0.11 |
| BalanceCascade w/ TreeSmote | 76.96±0.19 | 80.05±0.15 | 66.35±0.20 | 83.60±0.51 | 76.31±0.10 |

Table 20: Performance comparison (mean ± standard error) between ensemble methods and their combinations with TreeSMOTE across different imbalance levels (low, medium, high, and extreme) on F1-Score.

| Method | Low IRs | Medium IRs | High IRs | Extreme IRs | Average |
|---|---|---|---|---|---|
| SelfPacedEnsemble | 76.88±0.17 | 78.84±0.12 | 64.61±0.14 | 73.93±0.39 | 74.67±0.08 |
| SelfPacedEnsemble w/ TreeSmote | 76.73±0.18 | 78.98±0.11 | 64.78±0.14 | 74.31±0.40 | 74.74±0.09 |
| BalancedRandomForest | 75.85±0.18 | 75.84±0.12 | 58.87±0.14 | 55.49±0.34 | 70.46±0.09 |
| BalancedRandomForest w/ TreeSmote | 75.71±0.18 | 76.07±0.12 | 59.45±0.13 | 55.80±0.19 | 70.64±0.08 |
| RUSBoost | 75.80±0.20 | 76.55±0.11 | 60.45±0.16 | 62.28±0.32 | 71.61±0.09 |
| RUSBoost w/ TreeSmote | 76.14±0.19 | 77.03±0.13 | 60.90±0.18 | 62.25±0.38 | 71.99±0.10 |
| UnderBagging | 76.50±0.18 | 77.46±0.14 | 61.67±0.13 | 58.71±0.30 | 72.12±0.09 |
| UnderBagging w/ TreeSmote | 76.44±0.18 | 77.50±0.12 | 62.04±0.16 | 59.17±0.22 | 72.24±0.08 |
| BalanceCascade | 76.32±0.17 | 78.29±0.12 | 60.22±0.17 | 69.29±0.47 | 72.94±0.09 |
| BalanceCascade w/ TreeSmote | 76.74±0.18 | 77.97±0.12 | 60.33±0.16 | 70.05±0.43 | 73.07±0.09 |

Table 21: Performance comparison (mean ± standard error) between ensemble methods and their combinations with TreeSMOTE across different imbalance levels (low, medium, high, and extreme) on G-mean. .

| Method | Low IRs | Medium IRs | High IRs | Extreme IRs | Average |
|---|---|---|---|---|---|
| SelfPacedEnsemble | 77.01±0.17 | 79.23±0.12 | 65.68±0.14 | 75.74±0.34 | 75.24±0.08 |
| SelfPacedEnsemble w/ TreeSmote | 76.88±0.18 | 79.37±0.11 | 65.88±0.14 | 76.10±0.37 | 75.31±0.09 |
| BalancedRandomForest | 76.39±0.18 | 77.13±0.12 | 61.43±0.14 | 60.78±0.29 | 72.10±0.08 |
| BalancedRandomForest w/ TreeSmote | 76.26±0.18 | 77.34±0.12 | 61.88±0.12 | 61.07±0.17 | 72.25±0.08 |
| RUSBoost | 76.01±0.20 | 77.13±0.11 | 61.91±0.16 | 65.81±0.32 | 72.50±0.09 |
| RUSBoost w/ TreeSmote | 76.34±0.19 | 77.61±0.13 | 62.46±0.18 | 65.70±0.35 | 72.89±0.09 |
| UnderBagging | 76.68±0.18 | 78.11±0.14 | 63.35±0.12 | 63.36±0.27 | 73.17±0.09 |
| UnderBagging w/ TreeSmote | 76.62±0.18 | 78.17±0.12 | 63.70±0.15 | 63.82±0.20 | 73.29±0.08 |
| BalanceCascade | 76.39±0.17 | 78.64±0.12 | 61.37±0.17 | 71.73±0.44 | 73.54±0.09 |
| BalanceCascade w/ TreeSmote | 76.80±0.18 | 78.34±0.13 | 61.47±0.16 | 72.32±0.39 | 73.66±0.09 |

Table 22: Performance comparison (mean ± standard error) between ensemble methods and their combinations with TreeSMOTE across different imbalance levels (low, medium, high, and extreme) on MCC.

| Method | Low IRs | Medium IRs | High IRs | Extreme IRs | Average |
|---|---|---|---|---|---|
| SelfPacedEnsemble | 54.26±0.34 | 64.84±0.22 | 50.29±0.26 | 59.09±0.62 | 57.45±0.16 |
| SelfPacedEnsemble w/ TreeSmote | 54.00±0.36 | 65.13±0.20 | 50.79±0.25 | 59.35±0.65 | 57.59±0.16 |
| BalancedRandomForest | 53.73±0.35 | 61.73±0.19 | 43.75±0.23 | 33.53±0.41 | 52.60±0.15 |
| BalancedRandomForest w/ TreeSmote | 53.50±0.35 | 62.30±0.20 | 44.58±0.18 | 33.98±0.36 | 52.93±0.15 |
| RUSBoost | 52.39±0.40 | 60.89±0.19 | 44.26±0.28 | 40.09±0.49 | 52.51±0.17 |
| RUSBoost w/ TreeSmote | 53.00±0.38 | 61.85±0.22 | 44.88±0.29 | 39.51±0.54 | 53.14±0.17 |
| UnderBagging | 53.68±0.36 | 63.01±0.23 | 45.76±0.20 | 36.39±0.45 | 53.70±0.16 |
| UnderBagging w/ TreeSmote | 53.56±0.36 | 63.07±0.19 | 46.32±0.25 | 37.28±0.34 | 53.87±0.15 |
| BalanceCascade | 52.89±0.35 | 63.67±0.20 | 42.42±0.28 | 50.59±0.75 | 54.14±0.16 |
| BalanceCascade w/ TreeSmote | 53.72±0.36 | 63.25±0.23 | 42.38±0.27 | 51.11±0.63 | 54.33±0.17 |

## C.4 Comparision with SelfPacedEnsemble

Table 23: Comparison of the average performance of our method and SelfPacedEnsemble across various models in Table 1.

| Metrics | Method | Low IRs | Medium IRs | High IRs | Extreme IRs | Average |
|---------|--------|---------|------------|----------|-------------|---------|
| AUPRC | TreeSMOTE | 64.14±0.09 | 64.83±0.08 | 46.86±0.12 | 55.90±0.43 | 60.19±0.06 |
| | SelfPacedEnsemble | 71.55±0.21 | 71.65±0.16 | 54.66±0.21 | 64.86±0.88 | 67.39±0.12 |
| BAC | TreeSMOTE | 76.04±0.08 | 78.03±0.06 | 66.14±0.08 | 77.26±0.32 | 74.80±0.04 |
| | SelfPacedEnsemble | 77.21±0.18 | 80.83±0.16 | 69.66±0.20 | 85.03±0.52 | 77.51±0.10 |
| F1-Score | TreeSMOTE | 75.48±0.08 | 77.06±0.06 | 64.25±0.08 | 74.52±0.22 | 73.65±0.04 |
| | SelfPacedEnsemble | 76.88±0.17 | 78.84±0.12 | 64.61±0.14 | 73.93±0.39 | 74.67±0.08 |
| G-mean | TreeSMOTE | 75.64±0.07 | 77.29±0.06 | 64.69±0.08 | 75.22±0.23 | 73.93±0.04 |
| | SelfPacedEnsemble | 77.01±0.17 | 79.23±0.12 | 65.68±0.14 | 75.74±0.34 | 75.24±0.08 |
| MCC | TreeSMOTE | 51.57±0.15 | 61.45±0.10 | 46.62±0.13 | 57.52±0.34 | 54.32±0.08 |
| | SelfPacedEnsemble | 54.26±0.34 | 64.84±0.22 | 50.29±0.26 | 59.09±0.62 | 57.45±0.16 |
| Average | TreeSMOTE | 68.58±0.04 | 71.73±0.03 | 57.71±0.04 | 68.08±0.13 | 67.38±0.02 |
| | SelfPacedEnsemble | 71.38±0.09 | 75.08±0.06 | 60.98±0.08 | 71.73±0.20 | 70.45±0.05 |

Table 24: Comparison of the average performance of our method and SelfPacedEnsemble across various models in Table 3.

| Dataset | Method | AUPRC | BAC | F1-Score | G-mean | MCC | Average |
|---|---|---|---|---|---|---|---|
| optical_digits | TreeSMOTE | 91.05±0.24 | 95.26±0.17 | 94.53±0.13 | 94.62±0.13 | 89.29±0.27 | 92.95±0.08 |
| | SelfPacedEnsemble | 99.11±0.12 | 96.18±0.21 | 97.25±0.16 | 97.27±0.16 | 94.54±0.32 | 97.44±0.14 |
| satimage | TreeSMOTE | 43.12±0.48 | 75.92±0.35 | 70.40±0.31 | 72.06±0.25 | 46.96±0.42 | 61.69±0.15 |
| | SelfPacedEnsemble | 76.47±1.45 | 83.06±0.87 | 81.95±0.69 | 81.98±0.70 | 63.98±1.40 | 79.89±0.41 |
| pen_digits | TreeSMOTE | 93.92±0.17 | 97.42±0.09 | 96.01±0.06 | 96.16±0.06 | 92.43±0.13 | 95.19±0.05 |
| | SelfPacedEnsemble | 99.65±0.15 | 98.35±0.26 | 98.98±0.14 | 98.99±0.14 | 97.97±0.29 | 99.27±0.05 |
| sick_euthyroid | TreeSMOTE | 76.39±0.49 | 92.01±0.22 | 87.88±0.31 | 88.31±0.29 | 76.91±0.56 | 84.30±0.16 |
| | SelfPacedEnsemble | 89.24±1.23 | 92.47±0.93 | 93.09±0.72 | 93.11±0.72 | 86.23±1.43 | 91.41±0.29 |
| thyroid_sick | TreeSMOTE | 76.76±0.50 | 90.44±0.15 | 86.82±0.12 | 87.36±0.12 | 75.00±0.25 | 83.28±0.09 |
| | SelfPacedEnsemble | 96.24±0.08 | 91.24±0.25 | 93.94±0.28 | 94.05±0.29 | 88.13±0.58 | 91.17±0.17 |
| coil_2000 | TreeSMOTE | 11.03±0.09 | 55.35±0.13 | 54.43±0.10 | 55.09±0.09 | 10.70±0.17 | 37.32±0.05 |
| | SelfPacedEnsemble | 14.06±0.55 | 64.94±0.58 | 54.42±0.47 | 56.93±0.45 | 17.19±0.78 | 42.35±0.22 |
| wine_quality | TreeSMOTE | 25.20±1.07 | 67.09±0.44 | 65.97±0.49 | 66.43±0.48 | 33.04±0.95 | 51.55±0.29 |
| | SelfPacedEnsemble | 36.86±3.69 | 76.98±1.52 | 64.80±1.13 | 66.82±1.12 | 34.67±2.21 | 54.79±0.56 |
| letter_img | TreeSMOTE | 90.55±0.19 | 97.06±0.10 | 93.56±0.07 | 94.08±0.07 | 88.32±0.14 | 92.71±0.05 |
| | SelfPacedEnsemble | 99.35±0.17 | 96.87±0.20 | 98.14±0.13 | 98.16±0.13 | 96.32±0.25 | 98.45±0.13 |
| abalone_19 | TreeSMOTE | 3.66±0.50 | 50.82±0.29 | 50.98±0.42 | 51.35±0.56 | 2.80±1.13 | 31.92±0.26 |
| | SelfPacedEnsemble | 4.03±0.56 | 70.35±2.19 | 48.83±0.42 | 52.51±0.59 | 10.10±1.11 | 35.06±0.39 |
| isolet | TreeSMOTE | 82.29±0.18 | 91.50±0.11 | 91.22±0.08 | 91.24±0.08 | 82.48±0.17 | 87.75±0.05 |
| | SelfPacedEnsemble | 93.93±0.33 | 93.67±0.15 | 92.80±0.42 | 92.81±0.41 | 85.63±0.81 | 95.39±0.12 |

### C.5    Results on TabPFN

Table 25: Comparison of TreeSMOTE with classical oversampling methods across different imbalance levels (low, medium, high, extreme), with TabPFN as downstream classifier. Reported values are averaged over AUPRC, F1-Score, Balanced Accuracy, MCC, and G-mean.

| Oversampling Method | Imbalance Level | | | | |
| --- | --- | --- | --- | --- | --- |
| | Low | Medium | High | Extreme | Average |
| None | 71.72±0.09 | 75.65±0.05 | 64.19±0.07 | 76.14±0.27 | 71.93±0.04 |
| SMOTE | 69.85±0.09 | 71.54±0.07 | 56.46±0.11 | 56.80±0.30 | 66.72±0.05 |
| BorderlineSMOTE | 69.77±0.10 | 72.22±0.08 | 58.16±0.09 | 61.65±0.26 | 67.65±0.05 |
| SVMSMOTE | 69.93±0.09 | 72.69±0.08 | 59.17±0.10 | 62.05±0.31 | 68.10±0.05 |
| SMOTE-TomekLinks | 69.73±0.10 | 73.23±0.08 | 58.05±0.10 | 59.57±0.26 | 67.78±0.05 |
| TreeSMOTE | 70.75±0.09 | 72.99±0.06 | 58.81±0.10 | 60.42±0.30 | 68.31±0.05 |

Table 26: Comparison of TreeSMOTE and generative model-based oversampling methods on 10 datasets, with TabPFN as downstream classifier. Reported values are averaged over AUPRC, F1-Score, Balanced Accuracy, MCC, and G-mean.

| Dataset | TVAE | CTGAN | CTGANENN | DGOT | TabDDPM | TreeSMOTE |
| --- | --- | --- | --- | --- | --- | --- |
| optical_digits | 99.22±0.02 | 99.39±0.02 | 99.19±0.01 | 99.26±0.03 | 99.58±0.01 | 97.20±0.04 |
| satimage | 85.42±0.18 | 86.12±0.19 | 75.11±0.29 | 85.48±0.16 | 86.88±0.19 | 59.18±0.31 |
| pen_digits | 99.77±0.01 | 99.73±0.01 | 99.69±0.03 | 99.67±0.03 | 99.78±0.02 | 99.45±0.02 |
| sick_euthyroid | 86.53±0.17 | 88.44±0.11 | 65.73±0.34 | 90.84±0.13 | 91.69±0.13 | 80.99±0.16 |
| thyroid_sick | 87.09±0.14 | 89.72±0.14 | 62.65±0.54 | 89.21±0.16 | 91.89±0.09 | 78.30±0.12 |
| coil_2000 | 33.89±0.12 | 33.02±0.10 | 31.31±0.03 | 32.78±0.02 | 33.11±0.07 | 34.94±0.15 |
| wine_quality | 57.14±0.36 | 51.64±0.42 | 43.88±0.39 | 42.33±0.18 | 53.19±0.22 | 49.33±0.30 |
| letter_img | 98.93±0.05 | 99.12±0.04 | 98.78±0.03 | 99.75±0.01 | 99.67±0.01 | 97.16±0.02 |
| abalone_19 | 30.95±0.17 | 30.89±0.15 | 30.17±0.02 | 31.59±0.04 | 31.05±0.02 | 30.81±0.03 |
| isolet | 95.49±0.09 | 95.54±0.05 | 90.41±0.14 | 93.57±0.10 | 94.88±0.08 | 73.92±0.20 |

## C.6 Visualization

To provide an intuitive view of how TreeSMOTE affects the data distribution for real data, we visualize the sample distributions using t-SNE. Specifically, we select two datasets, *mfeat-zernike* and *Pulsar-Dataset-HTRU2*, and compare the original data distribution with those obtained after applying SMOTE and TreeSMOTE. For each dataset, we project the high-dimensional samples onto a two-dimensional space using t-SNE.

Figure 5 visualizes the results. For each dataset, we present three cases: the original data distribution, the distribution after applying SMOTE, and the distribution after applying TreeSMOTE. Compared with SMOTE, TreeSMOTE generates samples with less noise in the feature space, incurring class boundaries with a larger margin.

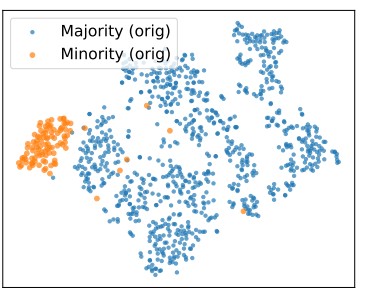
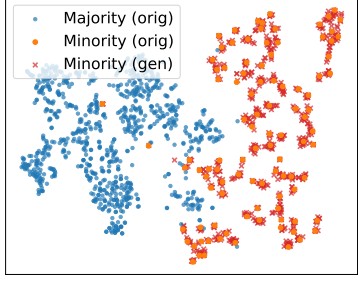
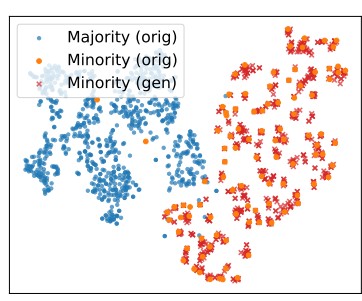

(a) Original distribution of the mfeat-zernike dataset.

(b) SMOTE-oversampled distribution of the mfeat-zernike dataset.

(c) TreeSMOTE-oversampled distribution of the mfeat-zernike dataset.

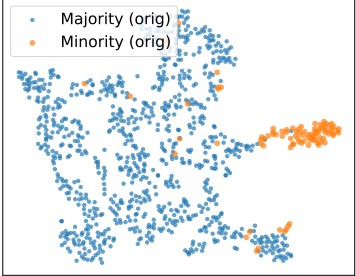
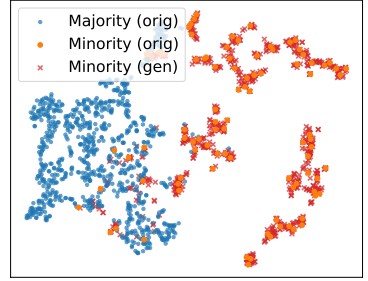
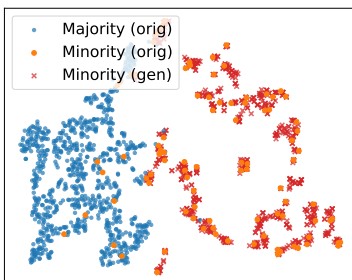

(d) Original distribution of the Pulsar-Dataset-HTRU2 dataset.

(e) SMOTE-oversampled distribution of the Pulsar-Dataset-HTRU2 dataset.

(f) TreeSMOTE-oversampled distribution of the Pulsar-Dataset-HTRU2 dataset.

Figure 5: t-SNE visualizations of data distributions on two datasets. For the *mfeat-zernike* dataset (Fig. 5a–5c), the original distribution, SMOTE, and TreeSMOTE results are presented. Compared with SMOTE, TreeSMOTE produces class boundaries with a larger margin. For the *Pulsar-Dataset-HTRU2* dataset (Fig. 5d–5f), the original data and the distributions after SMOTE and TreeSMOTE are shown. SMOTE tends to introduce noisy samples that may distort the data distribution, whereas TreeSMOTE better preserves the underlying data structure.

## C.7 Ablation Study

We conduct an ablation study to analyze the contributions of different components in our method. Specifically, we investigate from two perspectives: (1) the proposed structural gap versus Euclidean distance for finding nearest neighbors, and (2) using versus not using consistency filtering for synthetic sample elimination.

Table 27 presents the results of LightGBM as the downstream classifier across five evaluation metrics on extremely imbalanced datasets. The results show that using the proposed structural gap achieves an overall

Table 27: Ablation study of TreeSMOTE with LightGBM as the classifier on extremely imbalanced data.

| Proximity metric | Filtering | AUPRC | BAC | F1-Score | G-mean | MCC |
|---|---|---|---|---|---|---|
| Euclidean distance | No | 65.94 | 78.40 | 76.28 | 76.96 | 60.60 |
| Euclidean distance | Yes | 65.98 | 78.29 | 78.98 | 79.6 | 65.69 |
| Structural gap | No | 66.26 | **78.69** | 76.61 | 77.26 | 61.38 |
| Structural gap | Yes | **66.65** | 77.78 | **79.11** | **79.93** | **65.79** |

better performance than the Euclidean distance. Furthermore, applying consistency filtering improves results by excluding potentially noisy samples, thereby increasing the reliability of the generated synthetic data.

## C.8 Efficiency

To evaluate the computational efficiency of the oversampling methods, we compare their data augmentation time and CPU and GPU memory consumption. All experiments were conducted on a server equipped with an Intel Xeon Gold 5220 CPU and an NVIDIA A40 GPU. The software environment consists of Python 3.10.20, PyTorch 2.6.0 (CUDA 12.4), and scikit-learn 1.7.2. For the generative methods, we use the default training configurations provided in their official implementations, i.e., 300 epochs for TVAE, CTGAN, and CTGAN-ENN, 800 epochs for DGOT, and 30,000 diffusion steps for TabDDPM. For a given dataset, the number of synthetic samples generated remains consistent across all evaluated methods.

Table 28 details the computational efficiency of the evaluated data augmentation methods. The upper half of the table correspond to SMOTE-based methods evaluated on the 73 CLIMB data sets. TreeSMOTE is less efficient than SMOTE, BorderlineSMOTE, and SMOTE-TomekLinks, and is faster than SVMSMOTE. The high memory consumption is attributed to the tree training and consistency filtering for some big datasets. Focusing on the lower half of the table, which compares TreeSMOTE against generative frameworks on the 10 *imbalanced-learn* datasets, TreeSMOTE demonstrates a significant computational advantage in terms of execution time and memory consumption. In addition, it requires no GPU. The result establishes TreeSMOTE as a practical, out-of-the-box solution that does not necessitate specialized hardware.

Table 28: Time and memory consumption of data augmentation. The results are averaged over datasets. The upper and lower halves of the table correspond to the experiments in Table 1 and Table 3 in Section 4, respectively.

| Method | Time(s) | CPU Mem. (MB) | GPU Mem. (MB) |
|---|---|---|---|
| SMOTE | 0.16 | 20.90 | 0 |
| BorderlineSMOTE | 0.38 | 20.94 | 0 |
| SMOTE-TomekLinks | 3.01 | 24.83 | 0 |
| SVMSMOTE | 255.92 | 23.12 | 0 |
| TreeSMOTE | 93.07 | 956.91 | 0 |
| TVAE | 129.03 | 204.12 | 34.09 |
| CTGAN | 182.14 | 190.82 | 87.46 |
| CTGAN-ENN | 53.39 | 132.37 | 77.31 |
| DGOT | 108.40 | 281.21 | 285.04 |
| TabDDPM | 55.08 | 273.44 | 18.15 |
| TreeSMOTE | 1.03 | 29.26 | 0 |

