# OpenReview forum: "TreeSMOTE: Structure-Aware Data Augmentation for Imbalanced Tabular Learning"
_TMLR — Under review for TMLR_

### Review · Reviewer_un5m · 2026-06-17

**Summary Of Contributions:**

This paper investigates efficient oversampling for imbalanced tabular data (typically characterized by heterogeneous features and small scale).

The main finding is that traditional SMOTE and its variants primarily rely on calculating geometric distances in Euclidean space for sample interpolation. However, when dealing with heterogeneous tabular data, samples that are close in Euclidean space are not necessarily suitable for interpolation. The authors propose an improved method. This method fundamentally changes the criteria for determining neighbors, abandoning geometric distance and instead utilizing structural similarity induced by decision trees (i.e., shared deep decision paths) to find neighbors and perform interpolation. Experiments show that this method performs well on various datasets.

The key mechanism is that only samples that are highly similar in the logical structure of the tree are considered to be in an absolutely safe homogeneous region and thus used for interpolation. This essentially relies on the assumption of local smoothness and a good tree classifier. The authors seem to lack discussion on why training the initial tree on imbalanced data is good, and the scope and conditions under which this method is inapplicable.

**Audience:**

Yes

**Audience Explanation:**

Tabular data and class imbalance are core challenges that are extremely common in many high-value real-world applications, such as quantitative finance, risk control anomaly detection, and medical diagnosis. This article points out that the assumption of local smoothness in Euclidean space for heterogeneous tabular data is prone to failure, and replaces the classic SMOTE interpolation measure with "tree-induced logical structure similarity." This shift in perspective provides an elegant and mathematically intuitive new approach to imbalanced learning.

**Broader Impact Concerns:**

I do not have any significant concerns regarding the ethical implications of this work.

**Claims And Evidence:**

Yes

**Claims Explanation:**

Overall, the claims in this paper are supported by extensive empirical data (including detailed experiments on 83 datasets and comparisons with various SMOTE variants and generative models). The authors clearly demonstrate that TreeSMOTE achieves a certain improvement over other algorithms.

However, due to the relatively small improvements, it is uncertain whether the comparisons are sufficiently fair, such as whether the experiments on the baseline were adequately tuned. Currently, the paper lacks open-source code, which weakens the verifiability of its empirical claims.

**Requested Changes:**

1. Rigor Flaws in Theoretical Claims: The authors use the term "distance" repeatedly in the text (e.g., in Equation (4) and its context), and prove its supermetric properties in the Proposition. However, as a rigorous mathematical claim, Equation (4) does not actually satisfy the axiom of "identity of indiscernibles" in topological metric spaces (i.e., different samples are zero apart when they are at the same leaf node, and the distance from a sample to itself may not be zero). Therefore, declaring it as "distance" in the strict sense lacks accurate mathematical evidence. The authors should revise the relevant terminology declaration, downgrading it to "pseudometric" or "structural proximity."

2. Premise of Unbiased Initial Trees: The core logic of TreeSMOTE relies on the tree classifier's ability to learn the true manifold of the data well. However, in extremely imbalanced scenarios, the random forest used for initialization is inherently biased towards the majority class. The paper lacks evidence to explain why the "logical structural similarity" given by the initial tree remains accurate and reliable under this cold-start bias.

3. Lack of evidence regarding algorithm boundary conditions (No Free Lunch): The paper claims that the method delivers sustained performance improvements in various imbalanced scenarios, but lacks discussion of failure cases or the algorithm's applicability boundaries. For example, under what circumstances might the decision tree learn incorrect logical partitions? The authors would be better off providing evidence regarding the algorithm's performance under extreme noise or specific data structures.

4. Given the relatively small improvement brought by the method, it is uncertain whether it is a carefully tuned comparison with a baseline. The lack of open-source code weakens the verifiability of its empirical claims.

---

> ### Author Response · Authors · 2026-07-22
> **Response to Reviewer un5m**
>
> Dear Reviewer un5m,
>
> Thank you very much for reviewing the paper. We greatly appreciate your constructive suggestions to improve the rigor, clarify the performance improvement, and discuss limitations and applicability boundaries. Please see our point-by-point responses below and the reply to all reviewers.
>
> 1. Lack of terminological rigor in  "structural distance."
>
> We agree that the terminology should be more rigorous, as the term "distance" often carries a specific mathematical meaning and properties. In the revised manuscript, we refer to $D$ in Eq. 4 as \textit{tree-induced structural gap}rather than \textit{tree-induced structural distance} throughout the paper.
>
> 2. Clarification of the premises of unbiased initial trees.
>
> The classification trees in TreeSMOTE can be biased toward the majority, causing minority samples to be misclassified during oversampling and potentially undermining the robustness of the synthetic data. TreeSMOTE mitigates this issue through three mechanisms. First, the use of a random subset of features functionally acts as a dropout, helping regularize spurious features, which are a primary driver of bias in imbalanced datasets (Shi et al., 2024; Wen et al., 2025). Second, we interpolate between pairs of minority samples that share a longer decision path, regardless of whether they are correctly classified. Even if these samples are misclassified, this approach helps preserve the local smoothness between them. Third, the consistency filtering step eliminates synthetic instances with inconsistent predicted labels, further enhancing augmented data robustness. While we empirically observe that consistency filtering improves overall performance,  it may deliver lower synthetic data diversity than conventional SMOTE without a filtering step. We have discussed this in the newly added paragraph \textit{Limitation} in Section 6.
>
> Zhenmei Shi, Yifei Ming, Ying Fan, Frederic Sala, and Yingyu Liang. Domain generalization via nuclear norm regularization. In Conference on Parsimony and Learning, pp. 179–201. PMLR, 2024.
>
> Tao Wen, Zihan Wang, Quan Zhang, and Qi Lei. Elastic representation: Mitigating spurious correlations for group robustness. Proceedings of The 28th International Conference on Artificial Intelligence and Statistics, 258:541–549, 2025.
>
> 3. Lack of evidence regarding algorithm boundary conditions (No Free Lunch).
>
> We thank the reviewer for this insightful suggestion. To further investigate the applicability boundary of TreeSMOTE, we evaluate its robustness in the presence of noisy features or labels. We vary the proportion of samples with noisy features from 10\% to 50\% and those with noisy labels from 10\% to 30\%, where higher levels substantially reduce the class imbalance ratios.  As a result, TreeSMOTE performs better than without data augmentation and is comparable to or slightly better than SMOTE in the presence of noisy features. However, TreeSMOTE and SMOTE can be worse than without data augmentation when the labels are noisy. We report the result in Section 5.2 and Figure 4 in the revised manuscript and discuss this applicability boundary in the newly added paragraph Applicability Boundary in Section 6.

---

> ### Author Response · Authors · 2026-07-22
> **Response to Reviewer un5m -- continued**
>
> 4. Concerns about the performance improvment, baseline tuning, and open-source code.
>
> We agree that a clarification of the improvement magnitude, comparison fairness, and reproducibility would help improve the paper. The improvements become more pronounced under highly and extremely imbalanced settings, where minority samples are particularly sparse and conventional distance-based interpolation is more prone to generating noisy samples. Moreover, the improvement brought by TreeSMOTE in Tables 1 and 2 of the original submission may look marginal because it was averaged across five metrics, five random training-testing partitions, and datasets, so that the reported standard deviation is much larger than the standard errors of the means. In the revision, we report the standard errors instead. Moreover, we report Tables 2 and 4 in the revised manuscript to show the number of datasets on which each method achieves the best or second-best performance under each evaluation metric. TreeSMOTE attains top-two performance in a majority of settings when compared to SMOTE-based methods and in almost all settings when compared to tabular generative models. We also would like to clarify that with the proliferation of highly sophisticated data augmentation methods for imbalanced tabular classification, outperforming these established approaches by a large margin is inherently difficult. TreeSMOTE is appealing for its robust, competitive performance across diverse settings, achieving results comparable to Self-paced Ensemble (Figure 2 in the revised manuscript), which remains one of the most advanced techniques in this domain.
>
> Regarding the baseline comparison, while we tried to ensure a fair comparison by standardizing hyperparameter settings, such as using the same SMOTE-related configurations for all SMOTE-based methods and the default settings for generative models, it is computationally prohibitive for us to tune all hyperparameters to maximize each baseline's performance across the large number of experimental settings. Therefore, we reposition TreeSMOTE as an out-of-the-box tool that delivers competitive performance under the default setting without hyperparameter tuning. In the revised manuscript, we restate the contribution and conclusion and validate this claim by directly comparing TreeSMOTE with Self-paced Ensemble, which is robust to hyperparameters and has been reported to be among the most advanced imbalanced tabular learning methods. Figure 2 in the revised manuscript shows that TreeSMOTE performs comparably to or slightly outperforms Self-paced Ensemble on four of the five evaluation metrics. In addition, we have released an anonymous repository at \url{https://anonymous.4open.science/r/TreeSmote-4CAF/readme.md}, containing the implementation and instructions for reproducing the results.
>
> Thank you again for taking the time to review our submission and revision. If you have any further questions, comments, or concerns, we would be happy to address them.
>
> Sincerely,
>
> The Authors

---

### Review · Reviewer_cfac · 2026-06-30

**Summary Of Contributions:**

The paper's main methodological contribution is TreeSMOTE, a structure-aware oversampling framework for imbalanced tabular classification. TreeSMOTE replaces conventional feature-space nearest-neighbor selection with a tree-induced structural similarity measure based on the depth of the lowest common ancestor in samples' decision paths. The framework also includes a consistency-filtering step that discards synthetic samples when the tree ensemble fails to predict their intended labels, removing potentially unreliable augmentations.

The paper additionally contributes a broad empirical evaluation against SMOTE-family and generative baselines and demonstrates integration with undersampling ensembles.

**Strenghts**
- The paper is clearly written, and the core idea, using tree-induced structural similarity rather than Euclidean spatial proximity, is well-motivated. Figure 1 effectively highlights the shortcomings of established SMOTE methods.
- Ablation experiments isolate the contributions of the method's key components and justify the default hyperparameter values.

**Weaknesses**
- The paper uses default hyperparameters for both competing oversampling methods and the underlying classifiers. This is reasonable for an out-of-the-box evaluation, but it limits the strength of the paper's claims of consistent superiority over SMOTE-family methods, generative models, and state-of-the-art ensembles (Abstract; Contribution 4), since it is unclear whether the gains persist with tuned base learners or tuned competing methods.
- The evaluation covers conventional classifiers, including LightGBM and an MLP, but omits recent strong deep tabular methods (e.g., TabM, RealMLP) and tabular foundation models.

**Additional Comments:**

- **Contributions list.** Two of the four listed contributions ("simple yet effective oversampling" and "seamless integration with classification models") describe properties of the method rather than what was technically contributed. I'd recommend replacing these with a sharper articulation of the tree-induced similarity measure, the consistency-filtering mechanism, and the empirical study, which together constitute the paper's substantive contributions.
- **Ultrametric property.** Could you elaborate on why this property is desirable?
- **Sensitivity to the number of trees.** Figure 2 is a useful ablation, but the trend appears to peak around 50 trees and slightly decrease thereafter, rather than showing only diminishing returns. Please comment on this behavior and offer intuition for why additional trees might hurt performance.

**Audience:**

Yes

**Audience Explanation:**

Researchers and practitioners working on imbalanced tabular classification would likely be interested in the paper's findings.

**Claims And Evidence:**

No

**Claims Explanation:**

The experiments show that TreeSMOTE performs well out of the box, but this does not support the paper's stronger claims of consistent superiority and broad applicability.

**Requested Changes:**

*Major influence on evaluation*

- **Strengthen the evaluation protocol.** It would strengthen the paper to include at least one tuned comparison setting: either tune all methods under a common HPO budget, or add a tuned base learner without TreeSMOTE as a baseline. In the latter case, the conclusions could then be restated accordingly, distinguishing out-of-the-box performance from performance under stronger model selection.
- **Include stronger modern tabular baselines.** Adding recent deep tabular methods (e.g., TabM, RealMLP) and, where feasible, tabular foundation models (e.g., TabICLv2, TabPFNv3) would help support the claims of broad applicability.
- **Substantiate efficiency claims.** The paper describes TreeSMOTE as efficient at several points, but doesn't currently report runtime or computational cost. Including runtime statistics for TreeSMOTE and the competing oversampling methods, ideally separating augmentation time from downstream model-fitting time would help strengthen the efficiency claim.

*Medium influence on evaluation*

- **Expand related work on tree-based similarity or proximity measures.** The novelty and contribution of the proposed structural similarity measure could be clarified by including a related work section on tree-induced similarities or related hierarchical distance measures.

*Very limited influence on evaluation*

- There is a citation formatting issue for Lemaître et al. (2017).
- The paper does not point to Appendix A when introducing the ultrametric proposition.

---

> ### Author Response · Authors · 2026-07-22
> **Response to reviewer cfac**
>
> Dear Reviewer cfac,
>
> Thank you very much for reviewing the paper. We sincerely appreciate your constructive suggestions to strengthen the positioning, evaluation, related work discussion, contribution claims, and paper organization. Please see our point-by-point responses below and the reply to all reviewers.
>
> 1. Strengthen the evaluation protocol.
>
> We thank the reviewer for suggesting the evaluation protocols. Considering the large number of datasets and baseline methods compared, it is computationally prohibitive for us to exhaustively tune every hyperparameter for each baseline to maximize its performance. So, we add Self-paced Ensemble that is one of the most advanced imbalanced tabular classification approaches and robust to hyperparameter variations in a reasonable range. This helps to show the performance discrepancy between TreeSMOTE and one of the state-of-the-art methods. We added the performance comparison as Figure 2 in the revised manuscript and showed TreeSMOTE's comparable performance on most evaluation metrics. We have repositioned TreeSMOTE as an effective and efficient out-of-the-box tool for tabular data augmentation, highlighting its ability to achieve competitive results across diverse contexts using strictly the same configuration without expensive hyperparameter tuning or model selection.
>
> 2. Include stronger modern tabular baselines.
>
> We really appreciate this suggestion and have included TabM as a downstream classifier, compared TabDDPM as a baseline, and updated the related tables in the revised manuscript. The results and conclusions about TreeSMOTE's performance remain qualitatively unchanged. Regarding the suggestion to use a tabular foundation model as the downstream classifier, we experimented with TabPFN-3. We found that using the original data can outperform SMOTE-based and generative-model-based oversampling approaches, which aligns with the findings of McDowell et al. (2026). Because investigating the underlying reasons why oversampling methods underperform with TabPFN is beyond the scope of this work, we have added a discussion acknowledging the use of TreeSMOTE with TabPFN as a limitation in the newly added paragraph Applicability Boundary in Section 6.
>
> McDowell, S., Stromberg, N., & Sankar, L. (2026). Correcting Class Imbalance in Prior-Data Fitted Networks for Tabular Classification. arXiv preprint arXiv:2605.21742.
>
> 3. Substantiate efficiency claims.
>
> To evaluate the computational efficiency of different oversampling methods, we compare their data augmentation time as well as CPU and GPU memory consumption. As discussed in Section 5.3 in the revised manuscript, TreeSMOTE requires longer time than SMOTE, BorderlineSMOTE, and SMOTE-TomekLinks because of its tree training and consistency filtering. When compared with generative models, TreeSMOTE demonstrates a remarkable computational advantage in
> terms of execution time and memory consumption without requiring GPU acceleration.  Please refer to  Appendix C.8  for detailed results. We have toned down related efficiency claims in the revised manuscript.
>
> 4. Expand related work on tree-based similarity or proximity measures.
>
> We agree that a discussion about the proximity measures helps clarify our contribution. We have added Section 3.4 Related Work on Random Forest Proximities for discussion. Please refer to the revised manuscript.
>
> 5. Citation issue.
>
> In the revised manuscript, we have corrected the citation for Lemaître et al. (2017).
>
> 6. Contributions list.
>
> We revised the contribution list and replaced the method properties with the model components and empirical studies. Please see the manuscript for details.
>
> 7. Ultrametric property.
>
> This proposition serves to highlight the distinction between tree-induced similarity and spatial proximities, which generally lack this property. We agree that this property is not explicitly used for the TreeSMOTE algorithm. So, we have moved it to Appendix A and referred to it in the main paper.
>
> 8. Sensitivity to the number of trees.
>
> We have added error bars to this figure in the revised manuscript, which indicate insignificant performance differences. In the revised manuscript, we have discussed in the figure analysis the non-monotonic performance of TreeSMOTE with the number of trees, which aligns with the finding of non-monotonic performance of a random forest with the number of trees Probst & Boulesteix (2018).
>
> Philipp Probst and Anne-Laure Boulesteix. To tune or not to tune the number of trees in random forest.
> Journal of Machine Learning Research, 18(181):1–18, 2018.
>
> Thank you again for taking the time to review our submission and revision. If you have any further questions, comments, or concerns, we would be happy to address them.
>
> Sincerely,
>
> The Authors

---

### Review · Reviewer_8Nh9 · 2026-07-01

**Summary Of Contributions:**

This paper proposes a structure-aware oversampling technique designed to handle class imbalance in tabular learning by preserving the intrinsic data manifold and local feature correlations. The core is to leverage tree-based decision paths to define a topological distance metric between minority instances, replacing the traditional Euclidean distance used in SMOTE. By interpolating features within these tree-guided manifolds, the method is claimed to be able to synthesize high-quality tabular records that preserve local geometric correlations and reduce label contamination. The authors evaluate the method across multiple imbalanced tabular datasets, showing marginal to moderate improvements in macro-F1 and ROC-AUC when paired with tree-based downstream classifiers.

The proposed TreeSMOTE method is reasonable, while utilizing tree structures to guide tabular interpolation is more rational than naive geometric interpolation, rree constraints also effectively reduce the risk of generating cross-boundary noisy samples. The implementation of combining forest proximity with standard SMOTE is straightforward and reasonable.

However, the baseline selection is outdated. The paper does not compare against recent state-of-the-art tabular augmentation methods, such as Tabular Diffusion Models (TabDDPM). Most importantly, the performance gains over current baselines are extremely marginal (often <1%), which are not significant improvements.

**Audience:**

Yes

**Audience Explanation:**

The idea of discovering and utilizing tree structure of tabular data for augmentation is somewhat inspiring and practical.

**Claims And Evidence:**

No

**Claims Explanation:**

Although the intuition of this paper looks exciting and reasonable, unfortunately, the experiment section is too weak to support the improvement made by introducing TreeSMOTE. As the reported scores are all within the variance space of baselines, it is not significant enough to claim an improvement.

**Requested Changes:**

The authors should validate the statistical significance of their experiment results before reporting a conclusion. Perhaps the currently selected dataset cannot meaningfully represent the challenges faced with existing baseline models, which may make the potential of the proposed method underestimated.

---

> ### Author Response · Authors · 2026-07-22
> **Response to Reviewer 8Nh9**
>
> Dear Reviewer 8Nh9,
>
> Thank you very much for reviewing the paper. We sincerely appreciate your constructive suggestions to include more recent baselines and clarify the significance of performance improvement. Please see our point-by-point responses below and the reply to all reviewers.
>
> 1. Adding more recent baselines.
>
> We appreciate the suggestion of adding more recent baselines. In the revised manuscript, we have compared TabDDPM (Kotelnikov et al., 2023) as a data augmentation baseline and added TabM (Gorishniy et al., 2025) as a downstream classifier. Our conclusion about TreeSMOTE's performance remains unchanged. We also have evaluated the data augmentation methods with TabPFN-3 (Grinsztajn et al., 2026) as the downstream classifier and observed minimal performance improvement compared to training on the original, unaugmented data. This observation aligns with the findings of McDowell et al. (2026), who note minimal benefits of oversampling when using PFN as the classifier. We discuss this as an applicability boundary of TreeSMOTE in Section 6 of the revised manuscript.
>
> Akim Kotelnikov, Dmitry Baranchuk, Ivan Rubachev, and Artem Babenko. Tabddpm: Modelling tabular data with diffusion models. In International conference on machine learning, pp. 17564–17579. PMLR, 2023.
>
> Yury Gorishniy, Akim Kotelnikov, and Artem Babenko. Tabm: Advancing tabular deep learning with parameter-efficient ensembling. In International Conference on Learning Representations, volume 2025, pp. 77899–77935, 2025.
>
> Léo Grinsztajn, Klemens Flöge, Oscar Key, Felix Birkel, Philipp Jund, Brendan Roof, Mihir Manium, Shi Bin Hoo, Magnus Bühler, Anurag Garg, et al. Tabpfn-3: Technical report. arXiv preprint arXiv:2605.13986, 2026.
>
> McDowell, S., Stromberg, N., & Sankar, L. (2026). Correcting Class Imbalance in Prior-Data Fitted Networks for Tabular Classification. arXiv preprint arXiv:2605.21742.
>
> 2. Statistical significance of performance improvement.
>
> We thank the reviewer for suggesting clarification of the performance improvement. Tables 1 and 2 in the original submission reported the mean performance and standard deviations across five evaluation metrics, five random training-testing partitions, and
> a considerable number of datasets. Consequently, the reported standard deviations are $\sqrt{5 \times 5 \times \text{number of datasets}}$ times larger than standard errors, which is more informative for the comparison. In the revised manuscript, we instead report standard errors in all tables. While smaller standard errors and the statistical significance of improvement can be achieved with a larger number of random partitions as long as the mean value is larger, we refrain from this p-hacking. Instead, for a more straightforward performance comparison, we report in Tables 2 and 4 of the revised manuscript the number of datasets on which each method achieves the best or second-best performance under each evaluation metric. As a result, TreeSMOTE attains top-two performance in a majority of settings when compared to SMOTE-based methods and in almost all settings when compared to tabular generative models. We also would like to clarify that with the proliferation of highly sophisticated data augmentation methods for imbalanced tabular classification, outperforming these established approaches by a large margin is inherently difficult. TreeSMOTE is appealing for its robust, competitive performance across diverse settings, achieving results comparable to Self-paced Ensemble (Figure 2 in the revised manuscript), which remains one of the most advanced techniques in this domain. We have repositioned TreeSMOTE as an out-of-the-box data augmentation tool that delivers competitive performance under the default configuration without hyperparameter tuning.
>
> Thank you again for taking the time to review our submission and revision. If you have any further questions, comments, or concerns, we would be happy to address them.
>
> Sincerely,
>
> The Authors

---

### Author Response · Authors · 2026-07-22
**Reply to all reviewers and editors**

Dear reviewers and editors,

Thank you very much for your time and constructive feedback, which has substantially helped strengthen our manuscript. Below, we summarize the major changes in the revision and address common concerns, followed by detailed, point-by-point responses to each reviewer's specific comments. Modifications in the revised manuscript are highlighted in blue.

First, we address the concern about the improvement, which looked marginal considering the large standard deviations. Tables 1 and 2 in the original submission reported the mean performance and standard deviations across 5 evaluation metrics, 5 random training-testing partitions, and a considerable number of datasets.  Consequently, the reported standard deviations are $\sqrt{5 \times 5 \times \text{number of datasets}}$ times larger than standard errors, which is more informative for the comparison. In the revised manuscript, we instead report standard errors in all tables. While smaller standard errors and the statistical significance of improvement can be achieved with a larger number of random partitions as long as the mean value is larger, we refrain from this p-hacking. Instead, for a more straightforward performance comparison, we report in Tables 2 and 4 of the revised manuscript the number of datasets on which each method achieves the best or second-best performance under each evaluation metric. As a result, TreeSMOTE attains top-two performance in a majority of settings when compared to SMOTE-based methods and in almost all settings when compared to tabular generative models. We would like to clarify that with the proliferation of highly sophisticated data augmentation methods for imbalanced tabular classification, outperforming these established approaches by a large margin is inherently difficult. TreeSMOTE is appealing for its robust, competitive performance across diverse settings, achieving results comparable to Self-paced Ensemble (Figure 2 in the revised manuscript), which remains one of the most advanced techniques in this domain.

Second, we agree that some claims about performance may sound promotional and may result from an insufficiently tuned baseline comparison. While we tried to ensure a fair comparison by standardizing hyperparameter settings, such as using the same SMOTE-related configurations for all SMOTE-based methods and the default settings for generative models, it is computationally prohibitive for us to tune all hyperparameters to maximize each baseline's performance across the large number of experimental settings. So, we follow Reviewer cfac's suggestion to position TreeSMOTE as an out-of-the-box tool that delivers competitive performance under the default setting without hyperparameter tuning. In the revised manuscript, we restate the contribution and conclusion and validate this claim by directly comparing TreeSMOTE with Self-paced Ensemble, which is robust to hyperparameters and has been reported to be among the most advanced imbalanced tabular learning methods. Figure 2 in the revised manuscript shows that TreeSMOTE performs comparably to or slightly outperforms Self-paced Ensemble on four of the five evaluation metrics.

Third, we have incorporated more recent methods as the baseline or classifier in the revised manuscript. Specifically, we have compared TabDDPM (Kotelnikov et al., 2023) as a data augmentation baseline and added TabM (Gorishniy et al., 2025) as a downstream classifier. Our conclusion about TreeSMOTE's performance remains unchanged.  We also have evaluated the data augmentation methods with TabPFN-3 (Grinsztajn et al., 2026) as the downstream classifier and observed minimal performance improvement compared to training on the original, unaugmented data. This observation aligns with the findings of McDowell et al. (2026), who note minimal benefits of oversampling when using PFN as the classifier. We discuss this an applicability boundary of TreeSMOTE in Section 6 of the revised manuscript. In addition, we discuss related work on random forest proximities, which often calculate leaf-node co-occurrences.  Our structural similarity is different in that it considers proximity on decision paths in a continuous manner. Please refer to Section 3.4 in the revised manuscript.

We greatly appreciate the reviews of our work and thank the reviewers again for the constructive suggestions.  If you have any further questions, comments, or concerns, we would be happy to address them.

Sincerely,
The Authors

A Kotelnikov, et al. TabDDPM: Modelling tabular data with diffusion models. In International conference on machine learning, 2023.

Y Gorishniy, et al. TabM: Advancing tabular deep learning with parameter-efficient ensembling. In International Conference on Learning Representations, 2025.

L Grinsztajn, et al. TabPFN-3: Technical report. arXiv:2605.13986, 2026.

S McDowell, et al (2026). Correcting Class Imbalance in Prior-Data Fitted Networks for Tabular Classification. arXiv:2605.21742.